# Mapping the reduction in gross primary productivity in subarctic birch forests due to insect outbreaks

Per-Ola Olsson, Michal Heliasz, Hongxiao Jin, Lars Eklundh

Department of Physical Geography and Ecosystem Science, Lund University, Sölvegatan 12, S-223 62 Lund, Sweden.

*Correspondence to*: Per-Ola Olsson (per-ola.olsson@nateko.lu.se)

**Abstract.** It is projected that forest disturbances, such as insect outbreaks, will have an increasingly negative impact on forests with a warmer climate. These disturbance events can have a substantial impact on forests' ability to absorb atmospheric $CO_2$, and may even turn forests from carbon sinks into carbon sources; hence, it is important to develop methods to both monitor forest disturbances and to quantify the impact of these disturbance events on the carbon balance. In this study we present a method to monitor insect induced defoliation in a subarctic birch forest in northern Sweden, and to quantify the impact of these outbreaks on gross primary productivity (GPP). Since frequent cloud cover in the study area requires data with high temporal resolution and limits the use of finer spatial resolution sensors such as Landsat, defoliation was mapped with remote sensing data from the MODIS sensor with 250×250 m spatial resolution. The impact on GPP was estimated with a light use efficiency (LUE) model that was calibrated with GPP data obtained from eddy covariance (EC) measurements from five years with undisturbed birch forest and one year with insect induced defoliation. Two methods were applied to estimate the impact on GPP: (1) Applying a GPP reduction factor derived from EC measured GPP to estimate GPP loss, and (2) running a LUE model for both undisturbed and defoliated forest, and deriving the differences in modelled GPP. In the study area of 100 $km^2$ the results suggested a substantial setback to the carbon uptake: An average decrease in regional GPP over the three outbreak years (2004, 2012, and 2013) was estimated to 15±5 Gg C $yr^{-1}$, compared to the mean regional GPP of 40±12 Gg C $yr^{-1}$ for the five years without defoliation, i.e. 38 %. In the most severe outbreak year (2012), 76% of the birch forests were defoliated, and annual regional GPP was merely 50% of GPP for years without disturbances. The study has generated valuable data on GPP reduction, and demonstrates a potential for mapping insect disturbance impact over extended areas.

**Keywords:** Insect defoliation, subarctic mountain birch, MODIS, GPP, LUE

# 1 Introduction

It is estimated that forests account for half of the global terrestrial net primary productivity and act as important sinks of atmospheric $CO_2$ (Bonan 2008). Forests in the northern hemisphere contribute significantly to this sink, with the mid- and high latitude ecosystems as major contributors (Goodale et al. 2002; Kurz et al. 2008b). The high latitude forests are predicted to be among the ecosystems that are most strongly influenced by climate change (Kurz et al. 2008b); a warmer climate is likely to increase forest productivity (e.g. Nemani et al. 2003; Boisvenue & Running 2006), and result in higher uptake of $CO_2$ from the atmosphere. On the other hand, it is projected that the impact of forest disturbances will increase with a warmer climate (Seidl et al. 2014), and there are indications that disturbances such as wind, fires, and insect outbreaks have led to saturation of the carbon sink in European forests (Nabuurs et al. 2013). One important forest disturbance agent is insects; it is projected that the temporal and spatial dynamics, as well as the intensities and ranges of insect herbivore outbreaks will be influenced by global warming (Vanhanen et al. 2007; Battisti 2008; Jepsen et al. 2008; Netherer & Schopf 2010). These insect outbreaks can severely disturb forest ecosystems, and have a strong impact on carbon dynamics (Kurz et al. 2008a; Jepsen et al. 2009; Heliasz et al. 2011). Quantitative effects of insect outbreaks on the carbon balance are, however, not well known (Clark et al. 2010; Schäfer et al. 2010; Hicke et al. 2012), and insect outbreaks are generally excluded in large scale carbon modelling, which may result in overestimation of forests' ability to act as carbon sinks (Kurz et al. 2008b; Hicke et al. 2012). Consequently, it is important to develop methods both to monitor the spatial extent of insect outbreaks and to quantify the impact of these outbreaks on the carbon balance.

One alternative to estimate the impact on forest productivity is modelling: The impact of a large-scale outbreak of the mountain pine beetle (*Dendroctonus ponderosae* Hopkins) in British Columbia, Canada, was studied with a forest ecosystem model by Kurz et al. (2008a). Also the impact on the carbon balance of gypsy moth (*Lymantria dispar* L.) defoliation in New Jersey, USA, was modelled with both a canopy assimilation model (Schäfer et al. 2010) and a terrestrial biosphere model (Medvigy et al. 2012). The impact of spruce Budworm (*Choristoneura fumiferana* Clem.) outbreaks in eastern Canada were modelled by Dymond et al. (2010), and Landry et al. (2016) developed a Marauding Insect Module (MIM) in the Integrated Biosphere Simulator (IBIS) that enables simulation of insect outbreak for three insect functional types. Another alternative to quantify the influence of an insect outbreak on the carbon balance is to apply eddy covariance (EC) measurements: Brown et al. (2010, 2012) studied how a mountain pine beetle outbreak influenced net ecosystem productivity (NEP) in British Columbia, Canada; Clark et al. (2010, 2014) studied differences in NEE between undisturbed years and years with severe defoliation by the gypsy moth in New Jersey, USA; and Heliasz et al. (2011) estimated the reduction in net ecosystem exchange (NEE) during the growing season due to outbreaks of autumnal moth (*Epirrita autumnata* Borkhausen) and winter moth (*Operophtera brumata* L.) in northern Sweden in 2004. Even though not explicitly studied, there was gypsy moth defoliation of holm oak (*Quercus ilex* L.) present in a time-series of EC measurement in southern France (Allard et al. 2008). These methods generate valuable data on the impact of insect defoliation on the carbon balance; however, to quantify the total regional impact, data on the extent of defoliation events are required.

To generate wall-to-wall estimates of the disturbance effect on the carbon balance, remotely sensed data from satellites can be used. Several studies have demonstrated that satellite based remote sensing techniques can be applied to detect insect disturbances with high accuracy; see e.g. Wulder et al (2006); Adelabu et al. (2012) and Rullan-Silva et al. (2013) for reviews. In this paper we study outbreaks of autumnal moth and winter moth in subarctic mountain birch (*Betula pubescens*

*ssp. Czerepanovii* N.I. Orlova) forests in northern Sweden. These outbreaks often cover large areas, but are often followed by within-season recovery of the foliage in parts of the outbreak areas, which in combination with cloudy conditions can limit the possibility to map the outbreaks with remote sensing methods. Nevertheless, outbreaks of autumnal and winter moth have been mapped in northern Fennoscandia with high accuracy with Landsat data (Tømmervik et al. 2001; Babst et al. 2010). The low temporal resolution of Landsat (16 days revisit time) can, however, be a limitation; as an example, only

fractions of the area included in this study were visible in Landsat data during the peak of a severe outbreak in 2013. An alternative to Landsat data is coarse spatial resolution data from e.g. the moderate resolution imaging spectroradiometer (MODIS) sensor, which provides data with high (daily) temporal resolution and a spatial resolution of 250×250 m or coarser. MODIS derived Normalized Difference Vegetation Index (NDVI) have been used to map autumnal and winter moth outbreaks with high accuracy in northern Fennoscandia (Jepsen et al. 2009); and Olsson et al. (2016b) developed a method

for near real-time monitoring of insect induced defoliation that facilitates monitoring of refoliation later in the growing season.

Furthermore, there is a large body of research demonstrating that vegetation primary productivity can be estimated with remotely sensed data and a light use efficiency (LUE) approach (e.g. Prince 1991; Ruimy et al. 1994; Running et al. 2004; Xiao et al. 2004; Wu et al. 2010; McCallum et al. 2013; Gamon 2015). The LUE concept was introduced by Monteith (1972)

and Monteith & Moss (1977), suggesting that the primary productivity of plants has a strong linear relationship to the absorbed amount of photosynthetically active radiation (APAR), i.e. solar radiation in the spectral range 400–700 nm that is absorbed by the plant canopy. Since near-linear relationships between satellite derived vegetation indices and the fraction absorbed PAR (fAPAR) have been established (e.g. Asrar et al. 1984; Sellers 1987; Goward & Huemmrich 1992; Myneni & Williams 1994; Olofsson and Eklundh 2007), it is possible to create a LUE model driven by remote sensing data. Such a

LUE model could be applied for large-area estimates of the impact of forest disturbance on the uptake component of the carbon balance. Bright et al. (2013) utilized Landsat data to map bark beetle damage in northern Colorado, USA, and MODIS GPP data, which are based on a LUE model, to quantify the impact of the damage on GPP. However, to the knowledge of the authors, no previous study has utilized remote sensing data and developed a LUE model to monitor and quantify the impact of defoliating insects' outbreak on GPP.

In this study we utilized EC measured GPP to develop a LUE model, driven by MODIS derived NDVI, to quantify the regional impact on GPP of insect induced defoliation, and to map the spatial extent of the defoliation. Our main study objective was to compare GPP for years with insect damage (2004, 2012 and 2013) with GPP for years without insect damage (2007, 2009, 2010, 2011 and 2014) in the birch forest of a subarctic valley of northern Sweden. The analysis was achieved with two methods: (1) finding GPP for undisturbed forest and estimate the impact of an insect outbreak with a

common reduction factor derived from EC data; and (2) by applying a LUE model for both undisturbed and defoliated pixels and computing the differences.

## 2 Materials and methods

### 2.1 Study area

The study area was the mountain birch (*Betula pubescens ssp. Czerepanovii* N.I. Orlova) forests in a valley south-west of Abisko village (68.35N, 18.82E), and along the lake Torneträsk, as illustrated in Fig. 1 (green). The area is located in the subarctic zone in northern Sweden with lake Torneträsk at an altitude of 345 m.a.s.l. and with the highest mountains reaching 1 700 m.a.s.l. (Interact, 2016). These birch forests are infested by the autumnal moth (*Epirrita autumnata* Borkhausen) and the winter moth (*Operophtera brumata* L.) in time intervals of 9–10 years (Bylund 1995; Tenow et al.

2007). The first reported outbreaks by the autumnal moth in northern Fennoscandia are from mid-1800, and the winter moth has been observed in the northern parts of Fennoscandia since late 1800 (Tenow 1972). These insect outbreaks strongly influence the birch forests (Ammunét et al. 2015): Severe defoliation events may result in stem mortality, requiring decades of recovery (e.g. Tenow 1996; Tenow & Bylund 2000; Jepsen et al. 2013), and understorey vegetation can shift into more grass dominated communities (Karlsen et al. 2013; Jepsen et al. 2013). Root-associated fungal communities can change

(Saravesi et al. 2015), as well as chemical and physical properties of the soil (Kaukonen et al. 2013). A warmer climate, especially a lower frequency of years with extremely cold winters, as reported by Callaghan et al. (2010), strongly influences birch moth populations (Babst et al. 2010). The autumnal moth outbreaks have expanded into colder, more continental regions, and the winter moth has reached further to the north-east into areas where the autumnal moth previously dominated (Jepsen et al. 2008). The latest outbreaks in the study area occurred in 2004, with a documented reduction in carbon sink

strength of 89% at an EC tower located in birch forest (Heliasz et al. 2011), and in 2012 and 2013 (Bengt Landström, County administrative board of Norrbotten, pers. comm. 31.10.2013). These outbreak events were included in this study.

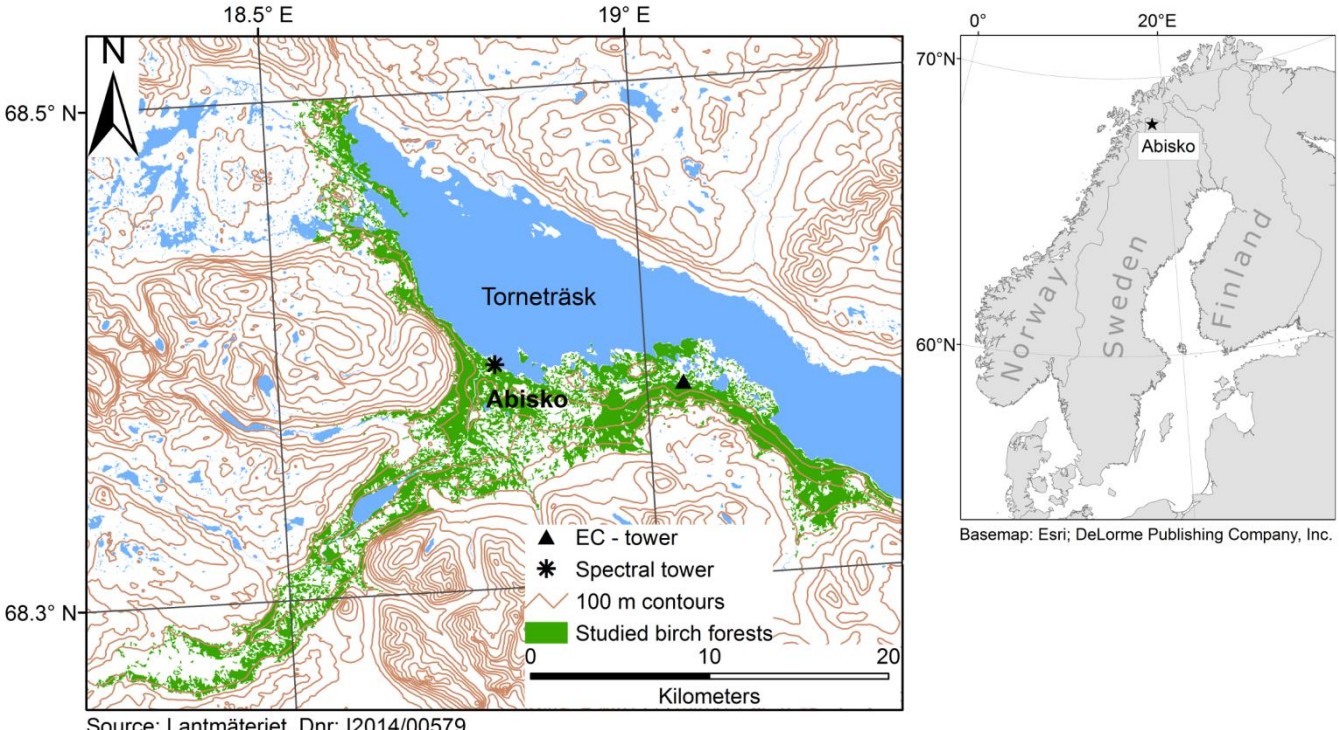

Source: Lantmäteriet, Dnr: I2014/00579

**Figure 1. The studied birch forest (green) along the south-west part of lake Torneträsk and in the valley to the south-west of Abisko village. The locations of the eddy covariance (EC) tower used to obtain GPP, and the spectral tower used to obtain fAPAR data are also shown. Reference system is SWEREF99 TM and latitude and longitude are in WGS84. Source of background map: Lantmäteriet (Dnr: I2014/00579).**

## 2.2 Data

### 2.2.1 Remote sensing data and smoothing of time-series

We used two Terra/MODIS satellite data products with eight days temporal resolution: (1) MOD09Q1 version 5, surface reflectance in the red and near infrared (NIR) bands, including quality assurance (QA) information, with 250×250 m spatial resolution, used mainly to derive NDVI (LPDAAC 2016a); and (2) MOD09A1 version 5, surface reflectance, as well as QA data, with 500×500 m spatial resolution (LPDAAC 2016b), utilized due to the product's more comprehensive QA data. NDVI was computed from the MODIS data as (Rouse et al. 1973; Tucker 1979):

$$NDVI = (NIR\text{-}red)/(NIR\text{+}red) \tag{1}$$

where *red* is reflectance in the red wavelength band, and *NIR* is reflectance in the near infrared wavelength band. We created time-series for the period 2000–2014 for all pixels in the study area and processed in TIMESAT ver. 3.2. TIMESAT is a software package used to reduce the influence of noise by fitting smoothed functions to time-series of data (Jönsson & Eklundh 2002, 2004). In this study we applied the same fittings and weights as in Olsson et al. (2016b): Double logistic functions were used to smooth the raw NDVI data and QA data from both MOD09Q1 and the more comprehensive QA-

flags in MOD09A1 were utilized to estimate the quality of the NDVI observations. In this study we use the term $NDVI_{DL}$ to refer to the smoothed time-series of NDVI.

### 2.2.2 Fraction of canopy absorbed PAR and relationships with NDVI

The fraction of PAR absorbed by the canopy ($fAPAR_{canopy}$) was measured at a spectral tower located in birch forest north-west from Abisko village (Fig. 1, black star). $fAPAR_{canopy}$ was obtained using the four-component method, i.e. measurements of incoming PAR above canopy, the total reflected PAR above the canopy, the transmitted PAR below the canopy, and the reflected PAR by the understorey vegetation and ground below the canopy. See Eklundh et al. (2011) for detailed information about the estimation of $fAPAR_{canopy}$. All PAR sensors were calibrated at the field site following the procedure by Jin & Eklundh (2015), and $fAPAR_{canopy}$ at solar noon time was calculated and used in the final analysis. $fAPAR_{canopy}$ data were available for the years 2010 and 2011.

Average $fAPAR_{canopy}$ over eight day periods, coinciding with the MODIS eight day periods, were computed, and an ordinary least squares (OLS) regression was performed to find the relationship between $fAPAR_{canopy}$ and $NDVI_{DL}$ Myneni & Williams (1994).The linear equation derived was used in the LUE model to obtain fAPAR from the double logistic fitted NDVI.

### 2.2.3 Eddy covariance and meteorological data

The EC tower is situated in the eastern part of the study area (Fig. 1, black triangle), and located near the crossing point of four nominal MODIS pixels with 250×250 m spatial resolution (Fig. 2). Vegetation in the four pixels is similar, with some open mires in the northeast pixel and a paved road crossing the two southernmost pixels. The tower's footprint is estimated to be about 200 m long, which is slightly smaller than a MODIS pixel. The prevailing wind directions are from the west and from the east, hence the main footprint of the EC tower is to the west and east from the tower where vegetation is most homogenous. Time-series of NDVI were extracted and mean values and standard deviations were computed for the four MODIS pixels to study if there were any larger deviations in the pixels' NDVI signals. In Figure 3, mean NDVI and standard deviation for the four pixels in the period 2010–2014 are shown. The low standard deviations indicate that there are minor differences in the NDVI signal between the pixels during the main growing season for both raw NDVI and $NDVI_{DL}$ both for years without disturbance and for outbreak years. Hence, we assume that a varying footprint of the EC tower due to varying wind directions and stability will have a limited influence on the EC measurements.

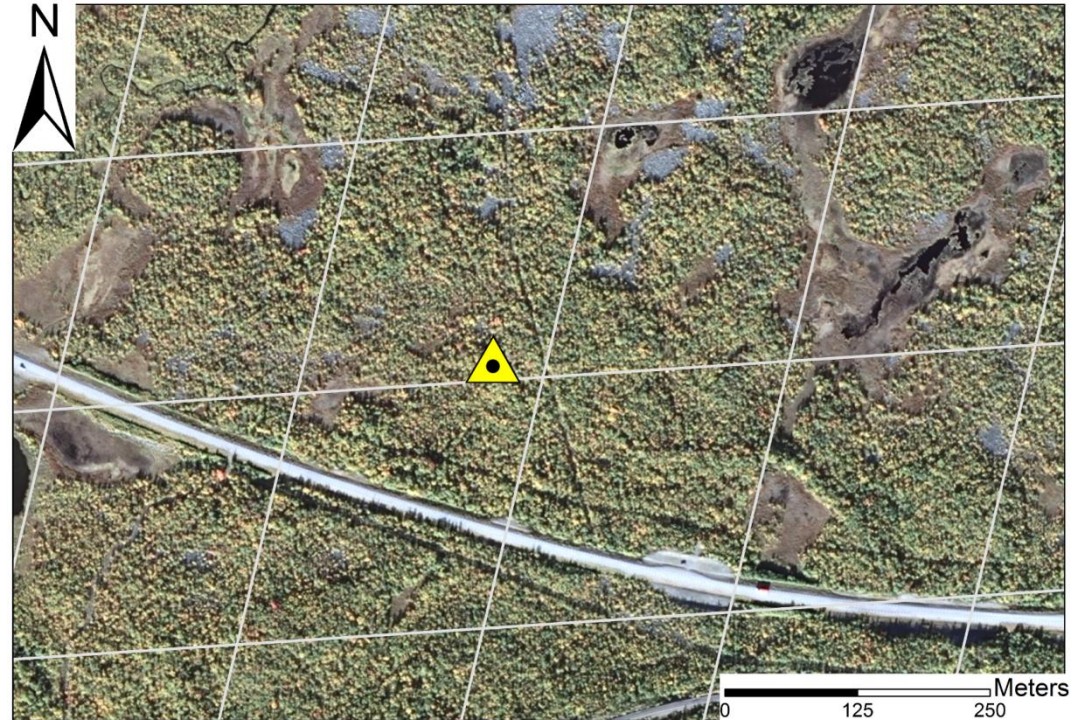

Source: Lantmäteriet Dnr: I2014/00579.

**Figure 2. The location of the eddy covariance (EC) tower (yellow triangle) near the crossing point of four nominal MODIS pixels with 250×250 m spatial resolution (white lines). Reference system: SWEREF99 TM. Lantmäteriet (Dnr: I2014/00579).**

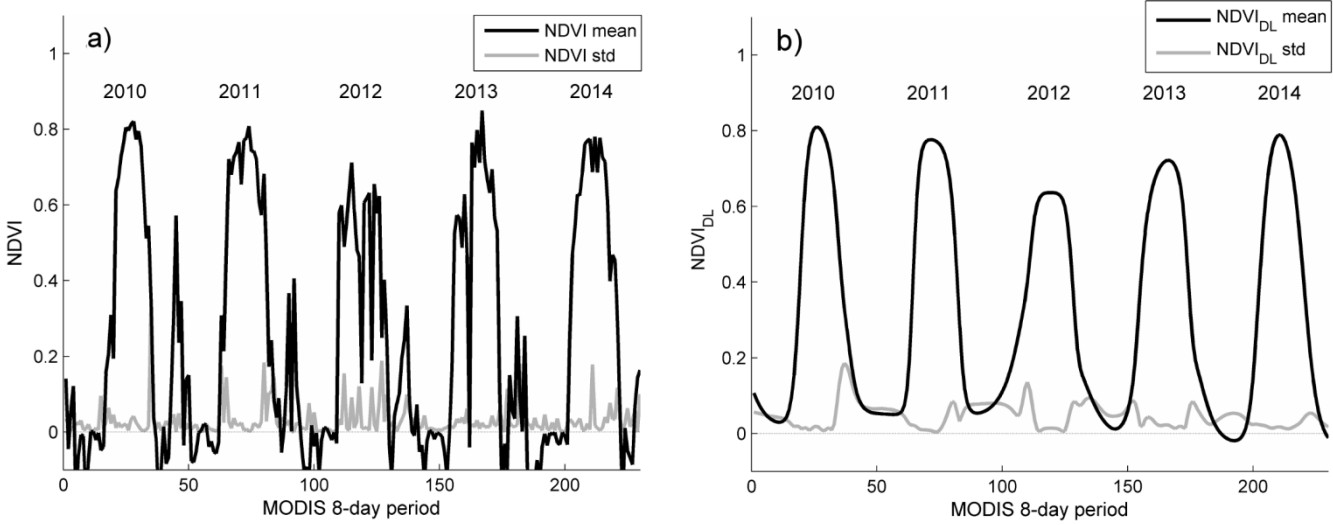

5     **Figure 3. Mean (black) and standard deviation (grey) of NDVI 2010–2014 for the four pixels around the eddy covariance (EC) tower. (a) is raw NDVI and (b) is NDVI fitted with double logistic functions in TIMESAT (NDVI$_{DL}$). 2012 and 2013 are years with insect outbreak. In 2013 the birch forest was refoliating later in the growing season. There is small secondary peak in raw NDVI (a) appearing each year. This peak appears during the winter when there is no vegetation in the study area, and is hence removed from the smoothed data (b).**

The EC measurements were made 8 m above ground, 3.3 m above canopy, using a 3-dimensional sonic anemometer (Metek USA-1; METEK Gmbh., Germany) and an open path infrared gas analyzer (Licor 7500, LI-COR Inc., USA). The system was operated with a frequency of 20 Hz, and data were recorded by a data logger (CR1000; Campbell Scientific, Inc., USA). Additional measurements of air temperature (Vaisala WXT510; Vaisala, Finland) and incoming photosynthetic flux density

(PPFD; JYP 1000, SDEC, France), used for flux partitioning and gap filling, were made at the tower. Data were obtained each year during the period May 1 to September 30, which is from before the start of the growing season (Karlsson et al. 2003) until late growing season; during the years included in this study GPP was approaching zero by the last week of September. For the years 2004 and 2013, temperature and PAR were obtained from Abisko scientific research station (ANS); comparisons between data from ANS and the EC tower showed small differences for the years when data were

available from both sources.

EC flux calculations were done with the EddyPro software ver. 5.2.1 (LI-COR Inc., USA). Gaps caused by bad weather conditions, bad EC measuring conditions, or short breaks in instrument functioning were filled with the online model: Eddy covariance gap-filling & flux-partitioning tool (http://www.bgc-jena.mpg.de/~MDIwork/eddyproc/). The main reasons for removing data were precipitation, as we used an open path gas analyser, and atmospheric conditions that did not fulfil the

turbulence conditions required. We considered the gap-filling approach suitable also for defoliated years since the gap-filling function is created based on data from short time windows, usually seven days, and hence, adjusts the fitting parameters for changing ecosystem conditions. A model from the same website was used to partition NEE into GPP and ecosystem respiration ($R_{eco}$). It was assumed that night time NEE is equal to night time $R_{eco}$. Accordingly, the accepted night-time data were fitted to the Lloyd and Taylor (1994) model based on air temperature. This model was also used to estimate $R_{eco}$ during

daytime conditions. GPP was estimated as the residual after subtracting $R_{eco}$ from the measured NEE. Details about gap filling and flux partitioning are described in Reichstein et al. (2005).

### 2.2.4 Land cover and elevation data

Land cover data were obtained from the Swedish mapping, cadastral, and land registration authority (Lantmäteriet; Dnr: I2014/00579). These land cover data are based on a classification of Landsat TM data, were updated in the year 2000 as a

part of the CORINE land cover project, and have a spatial resolution of 25×25 m (Lantmäteriet 2010). Birch forests in the study area were identified by extracting all pixels with broadleaved forest. Since birch is the dominating tree species with only a few sporadic individuals of other species (Sonesson & Lundberg 1974), all forests were considered to be birch. These data were used to calculate the fraction forest cover per MODIS pixel.

Elevation data were obtained from Lantmäteriet (National survey of Sweden) as a digital elevation model (DEM) with

50×50 m spatial resolution (Lantmäteriet; Dnr: I2014/00579). Mean elevation for each MODIS pixel was computed as the average altitude of all DEM pixels covered by a MODIS pixel. To adjust for altitudinal differences in temperatures across the study area, a mean summer temperature gradient of 0.5°C per 100 m (Josefsson 1990; Holmgren & Tjus 1996) was applied to the temperature data from the EC  tower.

**2.3 Light use efficiency model**

A LUE model with mean values of daily GPP in eight day intervals ($GPP_{lue}$) (g C m$^{-2}$ day$^{-1}$), corresponding to the time interval of the MODIS data, was developed as:

$$GPP_{lue} = \varepsilon \times fAPAR_{8day} \times PAR_{8day} \qquad (2)$$

where $\varepsilon$ (g C MJ$^{-1}$) is the light use efficiency, $fAPAR_{8day}$ is fAPAR for a MODIS eight day period derived from $NDVI_{DL}$, and $PAR_{8day}$ (MJ m$^{-2}$ day$^{-1}$) is mean daily PAR measured at the EC tower over the eight day period. The light use efficiency varies between vegetation types, and variability in meteorological conditions are accounted for through reductions factors for temperature and vapour pressure deficit (e.g. Field et al. 1995; Prince & Goward 1995; Potter et al. 1999; Turner et al. 2003). In this study the light use efficiency was computed as:

$$\varepsilon = \varepsilon_{max} \times f_{8day} \qquad (3)$$

where $\varepsilon_{max}$ (g C MJ$^{-1}$) (see Section 2.3.3) is the maximum efficiency applied in the model, and $f_{8day}$ is a reduction factor. We assumed that accounting for temperature only is sufficient in our study region, which is supported by Bergh et al. (1998) and Lagergren et al. (2005). Two models were created to describe $f_{8day}$, as in Lagergren et al. (2005): One model for the first part of the growing season and one model for the second part of the growing season.

**2.3.1 First part of the growing season**

During the first part of the growing season, covering May to late June, $f_{8day}$ depended on growing degree days (GDD) and frost events, where GDD was computed with a base temperature of 5°C, following Senn's et al. (1992) method applied to mountain birch development in northern Finland:

$$GDD_t = \begin{cases} GDD_{t-1} & , T_{mean8} \leq 5 \\ GDD_{t-1} + T_{mean8} - 5 & , T_{mean8} > 5 \end{cases} \qquad (4)$$

where $T_{mean8}$ (°C) is the mean temperature for a MODIS eight day period. The reduction factor was computed as:

$$f_{8day} = \begin{cases} 1 & , GDD_t \geq GDD_{thres} \\ 1 - \dfrac{GDD_{thres} - S_{GDD}}{GDD_{thres} + S_{GDD}} & , GDD_t < GDD_{thres} \end{cases} \qquad (5)$$

where $GDD_{thres}$ (see Section 2.3.3) is a threshold applied to decide when temperature and frost events no longer influence $\varepsilon$, in a similar fashion as Bergh et al. (1998) and Lagergren et al. (2005). $S_{GDD}$ is a reduction factor influenced by GDD and frost events and computed as:

$$S_{GDD} = \frac{GDD_t}{1 + P_{frost}} \qquad (6)$$

where $P_{frost}$ is a reduction factor controlled by frost events and computed as:

$$P_{frost} = \begin{cases} 0 & , T_{min8} \geq -3 \\ \dfrac{0.05 \times (-3 - T_{min8})}{5} & , -8 \leq T_{min8} < -3 \\ 0.05 & , T_{min8} < -8 \end{cases} \tag{7}$$

where $T_{min8}$ (°C) is the lowest temperature during a MODIS eight day period.

### 2.3.2 Second part of the growing season

In the second part of the growing season, covering late June to September, $f_{8day}$ is controlled by mean temperature only as:

$$f_{8day} = \begin{cases} 1 & , T_{mean8} \geq T_{thres} \\ \dfrac{T_{mean8}}{T_{thres}} & , T_{mean8} < T_{thres} \end{cases} \tag{8}$$

where $T_{thres}$ (°C) (see Section 2.3.3) is a temperature factor for controlling the influence of the eight day mean temperature during the second part of the growing season.

### 2.3.3 LUE model optimization

The LUE model was optimized to find three factors: (1) the GDD threshold ($GDD_{thres}$), (2) the temperature factor ($T_{thres}$), and

10 (3) the period to change from the first to the second seasonal model. These were found by minimizing the root mean square error (RMS) and maximizing $R^2$, based on $GPP_{lue}$ and daily mean values of GPP from the EC tower over MODIS eight day periods ($GPP_{EC}$). To compute $\varepsilon_{max}$, the mean value of the light use efficiency for all MODIS periods with maximum efficiency i.e. $f_{8day} = 1$ was calculated, where the efficiency was computed as:

$$\varepsilon_{max} = \frac{GPP_{EC}}{fAPAR_{8day} \times PAR_{8day}} \tag{9}$$

15 where $GPP_{EC}$ was derived from the EC tower. Two $\varepsilon_{max}$ values were computed: one including data from the five years (2007, 2009, 2010, 2011 and 2014) with undisturbed birch forest, and one ($\varepsilon_{max, def}$) for the year 2012 with insect defoliation. No data were available from 2008 due to equipment failure, and in 2013 the measurements were disturbed by larvae climbing the equipment.

### 2.3.4 LUE model uncertainty

20 A Monte Carlo approach was applied to evaluate the uncertainty of the LUE model by creating sets with 100 parameter values each for $\varepsilon_{max}$ and slope and intercept derived from the OLS regression between $fAPAR_{canopy}$ and $NDVI_{DL}$. The standard deviation of $\varepsilon_{max}$ was estimated from all MODIS periods with maximum efficiency, as described in 2.3.3, and a 95% confidence interval for the regression line was estimated. The different sets of parameters were created randomly from a uniform distribution, and the Monte Carlo simulation was run for all possible combinations of parameter values for the five

years with undisturbed forests and over 15 sets of 100 MODIS pixels with birch forest. Mean and standard deviation of LUE modelled GPP were estimated from these simulations.

## 2.4 Identifying MODIS pixels with defoliated birch forest

Defoliated MODIS pixels were identified for the three years with insect outbreaks with a near real-time monitoring method based on Kalman filtering and cumulative sums (Olsson et al. 2016b). The method identifies a seasonal trajectory of NDVI representing birch forest during a year without disturbances, called *stable season*. A Kalman filter (Kalman 1960) is applied to the raw NDVI observations from the year of study and deviations from the stable season are computed. A cumulative sum (CUSUM) filter (Page 1954) is applied to these deviations, and a pixel is classified as defoliated when the cumulative sum of deviations reaches a given threshold. In a near real-time application the stable season can only be derived from years prior to the year of study. In this study we modified the method so that the stable season was derived from all years with available data. For high detection accuracy, the method requires that a MODIS pixel is covered by at least 50% forest. Hence, based on the land cover data from Lantmäteriet, forest in pixels with lower forest cover were excluded, resulting in 100 $km^2$ of the totally 125 $km^2$ birch forest in the study area being included; the mean forest cover was 80% per MODIS pixel. The method detected 74% of the defoliated sampling areas in the study area with a misclassification of undisturbed areas of 39% (Olsson et al. 2016b).

## 2.5 Annual GPP loss due to insect defoliation

GPP for years without insect defoliation was estimated for all pixels by applying the LUE model and computing the mean value for the five years without insect outbreak, and with data available from the EC tower. The eight day average of incoming PAR ($PAR_{8day}$), measured at the EC tower, was assumed to be valid for all pixels in the study area, which was also suggested by comparisons between PAR measured at the EC tower and ANS.

Two methods were applied to study the reduction in annual GPP due to the insect outbreaks: (1) a method based on a reduction factor derived from the EC data from 2012 when the birch forest in the footprint of the tower was severely defoliated, and no refoliation occurred. This reduction factor was applied to all pixels in the study area, and (2) a method where the LUE model was applied to all defoliated pixels with $\varepsilon_{max, def}$ computed for defoliated growing seasons, and where the loss in GPP was computed as the difference between undisturbed and defoliated years.

### 2.5.1 Method 1 - GPP reduction factor

The fraction of the measured annual GPP at the EC tower that was lost due to the insect outbreak in 2012 was computed as:

$$GPP_{redfact} = 1 - GPP_{defoliated}/GPP_{undisturbed} \qquad (10)$$

Where $GPP_{redfact}$ is the reduction factor and $GPP_{defoliated}$ is annual GPP from the EC tower in 2012. $GPP_{undisturbed}$ is GPP from the tower representing a year without disturbances and computed as the mean of annual GPP for the five years without disturbances.

The reduction in annual GPP was computed for each pixel by applying the reduction factor to GPP for undisturbed years and multiplying with the area forest cover in the pixel. The same reduction factor was applied to all years with insect defoliation. The total impact of the defoliation was computed as the sum of GPP loss for all defoliated pixels in the study area, and for each year with insect outbreak.

### 2.5.2 Method 2 - LUE model for defoliated pixels

The LUE model, modified to model growing season with defoliation, was applied to all defoliated pixels in the study area to estimate annual GPP for each year with defoliation. Derivation of $\varepsilon_{max, def}$ was done with the same method as $\varepsilon_{max}$, but only data from one year with insect outbreak (2012) were available to estimate $\varepsilon_{max, def}$, and to evaluate the performance of the defoliation LUE model. For each year with insect outbreak, the regional reduction in GPP was computed by summing, over all pixels identified as defoliated, the difference between GPP for years without outbreak and GPP for this specific outbreak year.

### 2.5.3 Influence of refoliation

We also studied how recovering foliage later in the growing season influenced the two methods. The assumption was that recovering foliage would result in slightly higher $NDVI_{DL}$ values, which would enable Method 2 to capture the refoliation and hence, estimate GPP losses more accurately. All pixels that were detected as defoliated were classified as refoliated or non-refoliated with the defoliation monitoring method. The differences between GPP loss derived with Method 1 and Method 2 were computed as *GPP loss method 1 - GPP loss method 2*. Finally, the mean differences for refoliated and non-refoliated pixels were derived.

### 3 Results

### 3.1 Correlation between fAPAR and NDVI

There was a strong linear relationship between eight day mean values of $fAPAR_{canopy}$ and $NDVI_{DL}$ for $NDVI_{DL}$ values $\geq 0.4$ (Fig. 4). The influence of observations with $NDVI_{DL}$ values $< 0.4$ and with $f_{8day} > 0$ was small. For the years with data available from the EC tower 8% of the eight day periods had $NDVI_{DL} < 0.4$ and $f_{8day} > 0$ in the MODIS pixels surrounding the tower. For these time periods average $f_{8day}$ was 0.068. Hence, an OLS regression equation was calculated with $NDVI_{DL}$ values $\geq 0.4$ to model the relationship between $fAPAR_{8day}$ and $NDVI_{DL}$. This resulted in an $R^2$ of 0.81 and the relationship:

$$fAPAR_{8day} = -0.05 + 0.60 \times NDVI_{DL} \qquad (11)$$

The 95% confidence intervals for slope and intercept applied in the Monte Carlo simulation to estimate the LUE model's uncertainty were -0.05±0.18 (intercept) and 0.60±0.11 (slope).

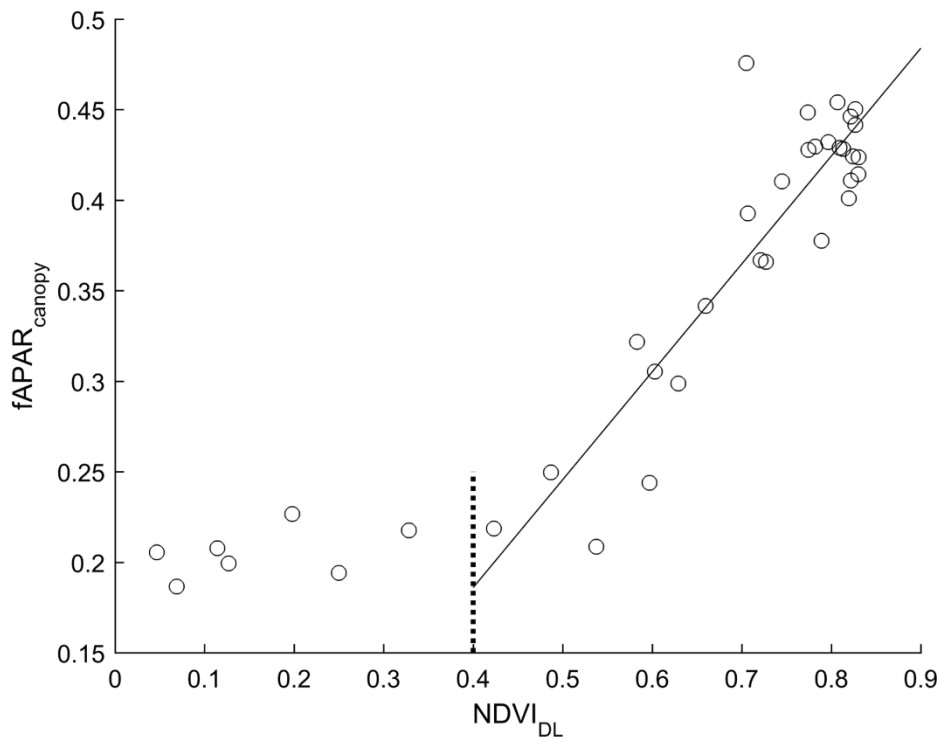

**Figure 4. Correlation between ground measured fraction of absorbed PAR by the canopy (fAPAR$_{canopy}$) and MODIS derived NDVI smoothed with a double logistic function in TIMESAT (NDVI$_{DL}$) in eight days intervals. Only NDVI$_{DL}$ values ≥ 0.4 were included in the OLS regression resulting in the black line. R$^2$ = 0.81 and N = 29.**

5    ### 3.2 Light use efficiency

Optimization resulted in a *GDD$_{thres}$* of 32 growing degree days (Fig. 5, left) and a *T$_{thres}$* of 8°C (Fig. 5, middle). The optimal period to change the model for *f$_{8day}$* was after MODIS period 23 i.e. the last week of June (Fig. 5, right).

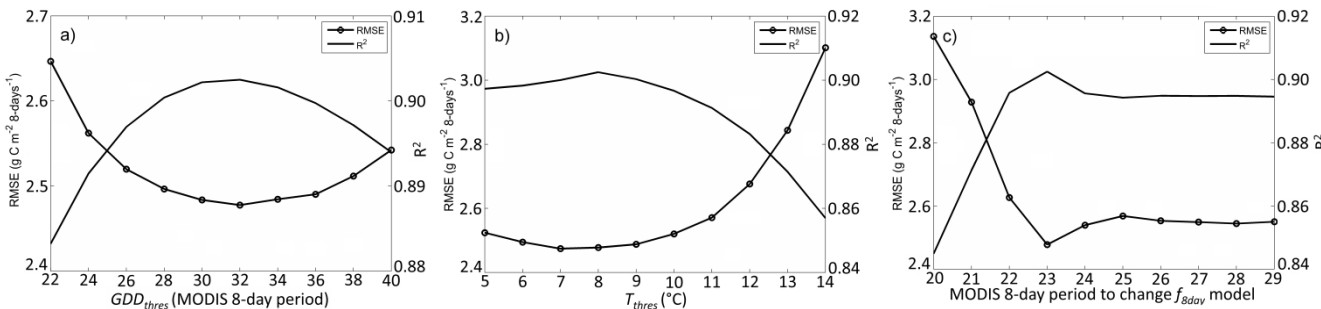

**Figure 5. Influence on RMSE and R$^2$ of GDD$_{thres}$ (a), T$_{thres}$ (b), and the optimal period to change from the first to the second f$_{8day}$**
10   **model (c). RMSE is computed from mean of daily GPP over eight day periods.**

Light use efficiency for years with no disturbance and with *f$_{8day}$* = 1 (black line with error bars in Fig. 6) gave an *ε$_{max}$* of 1.85±0.36 g C MJ$^{-1}$ (±1 standard deviation), resulting in the following LUE model:

$$GPP_{lue} = 1.85 \times f_{8day} \cdot (-0.05 + 0.60 \times NDVI_{DL}) \times PAR_{8day} \qquad (12)$$

The correlation between $GPP_{EC}$ and $GPP_{lue}$ was strong with $R^2 = 0.90$ (Figure 7). The intercept is -0.11 and the slope is 1.01 indicate that the LUE model performs well for years without outbreaks. The low GPP observations with several zero values for LUE modelled GPP are from May, before budburst for the birch forest. These low GPP values have little influence on annual GPP. The Monte Carlo simulation resulted in an estimated standard deviation of 30% of the mean annual GPP. Hence, all annual GPP values derived from the LUE model are given with a standard deviation of 30% of annual GPP.

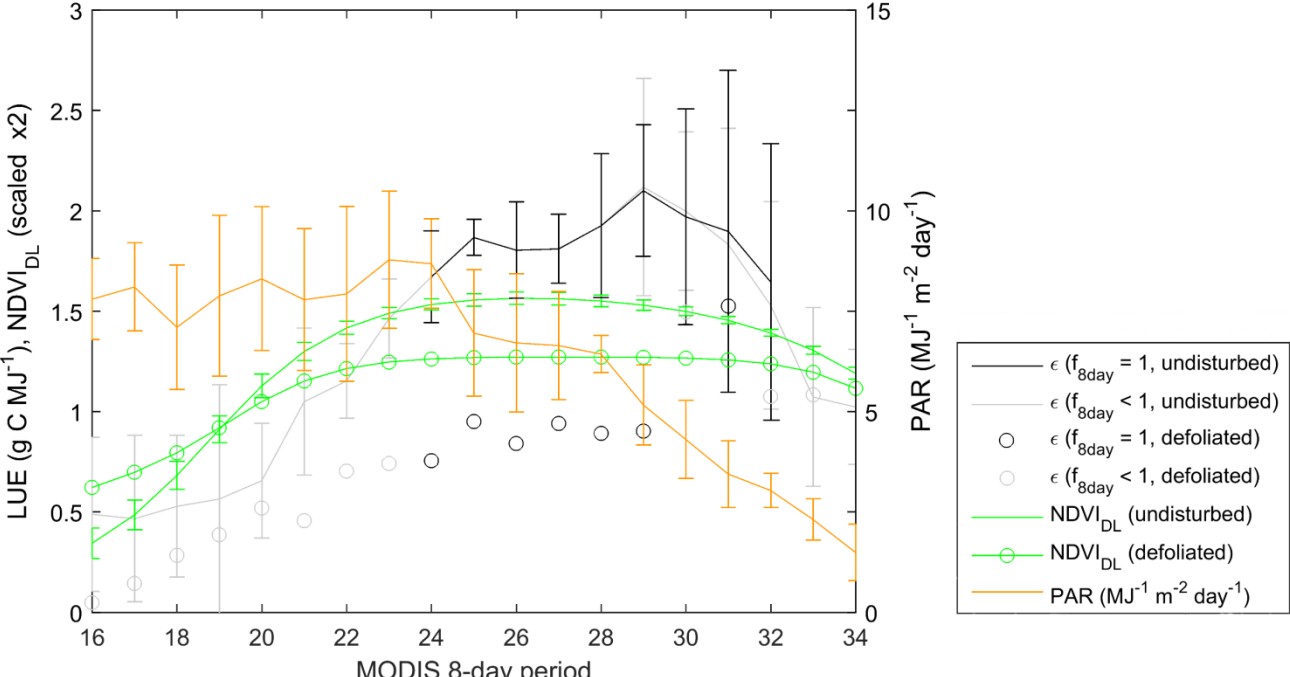

Figure 6. Light use efficiency (ε), NDVI fitted with double logistic functions (NDVI$_{DL}$) scaled ×2 (green), and PAR (orange) for the six years with data from the EC tower. Black lines with error bars and black circles are the light use efficiency values included when ε$_{max}$ and ε$_{max, def}$ were computed for undisturbed and defoliated years, respectively. The error bars are symmetric and one standard deviation higher or lower than the mean values.

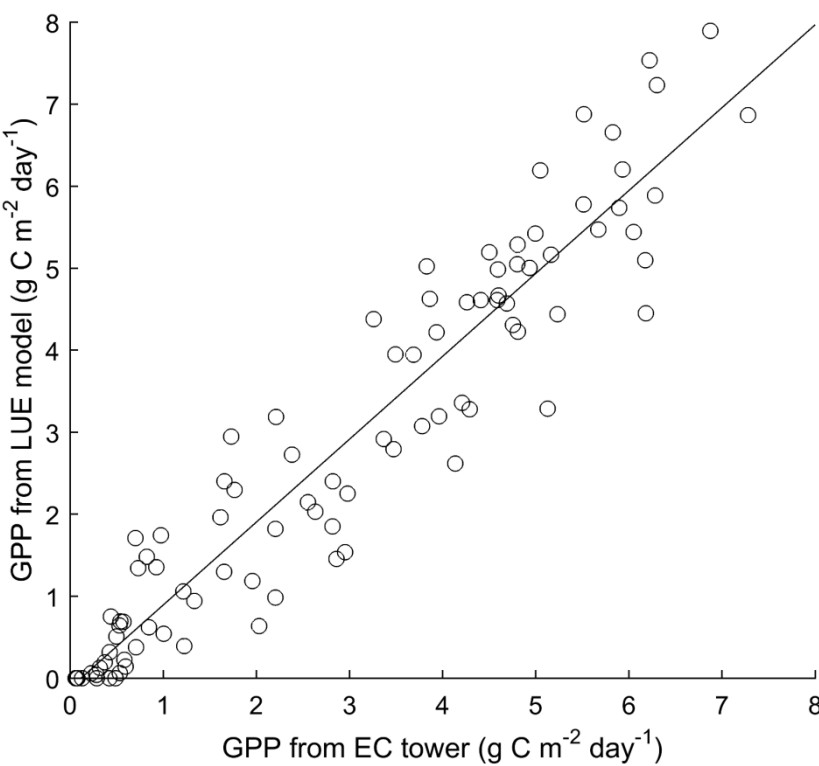

**Figure 7. Correlation between GPP from the EC tower and LUE modelled GPP for the five years with undisturbed forests. $GPP_{lue} = -0.11 \times 1.01 GPP_{EC}$, $R^2 = 0.90$ and N = 95.**

### 3.3 Impact of insect outbreaks on annual GPP

5  **3.3.1 Reduction factor and LUE model applied to quantify loss in GPP**

*Method 1 - reduction factor*

GPP measured from the EC tower and the five years with available data (Table 1) resulted in a mean annual GPP of 440 g C m$^{-2}$ yr$^{-1}$. During the outbreak in 2012 annual GPP was 180 g C m$^{-2}$ yr$^{-1}$, which resulted in a reduction in GPP compared to undisturbed conditions of 59%. Hence, a reduction factor of 0.59 was applied to quantify the impact of the

10  insect outbreak on GPP.

**Table 1. Annual GPP derived from the EC tower for the five years without insect outbreak and the year 2012 with insect outbreak.**

| | Years without insect outbreak | | | | | Outbreak |
|---|---|---|---|---|---|---|
| Year | 2007 | 2009 | 2010 | 2011 | 2014 | 2012 |
| **GPP** (g C m$^{-2}$ yr$^{-1}$) | 450 | 530 | 370 | 400 | 450 | 180 |

*Method 2 - LUE model for defoliated pixels*

The correlation between $GPP_{EC}$ and $GPP_{lue}$ for the year with defoliation (2012) and data available from the EC tower was weaker than for years without disturbances, with an $R^2$ of 0.83 (Fig. 8). The figure, with an intercept of -0.54 and a slope of 1.25, indicates that the LUE model underestimates GPP for lower values. The light use efficiency for the MODIS eight day periods with $f_{8day} = 1$ (black circles in Figure 6) gave an $\varepsilon_{max, def}$ of $0.98\pm0.25$ g C MJ$^{-1}$ ($\pm1$ standard deviation), resulting in the following LUE model for defoliated pixels:

$$GPP_{lue, defoliated} = 0.98 \times f_{8day} \cdot (-0.05 + 0.60 \times NDVI_{DL}) \times PAR_{8day} \tag{13}$$

In Figure 6, $NDVI_{DL}$ has higher values in the year with defoliation compared to undisturbed years in May (period 16-18). These high $NDVI_{DL}$ values are due to poor fitting of the double logistic function during winter and early spring in 2012 (see Figure 3, where $NDVI_{DL}$ increases earlier in 2012 compared to the other years). The impact on the result is, however, small, since these eight periods are in the early part of the growing season when $f_{8day}$ is zero.

The Monte Carlo simulation resulted in an estimated standard deviation of 35% of the mean annual GPP for years with defoliation. Hence, all annual GPP losses estimated with model 2 are given with a standard deviation of 35%.

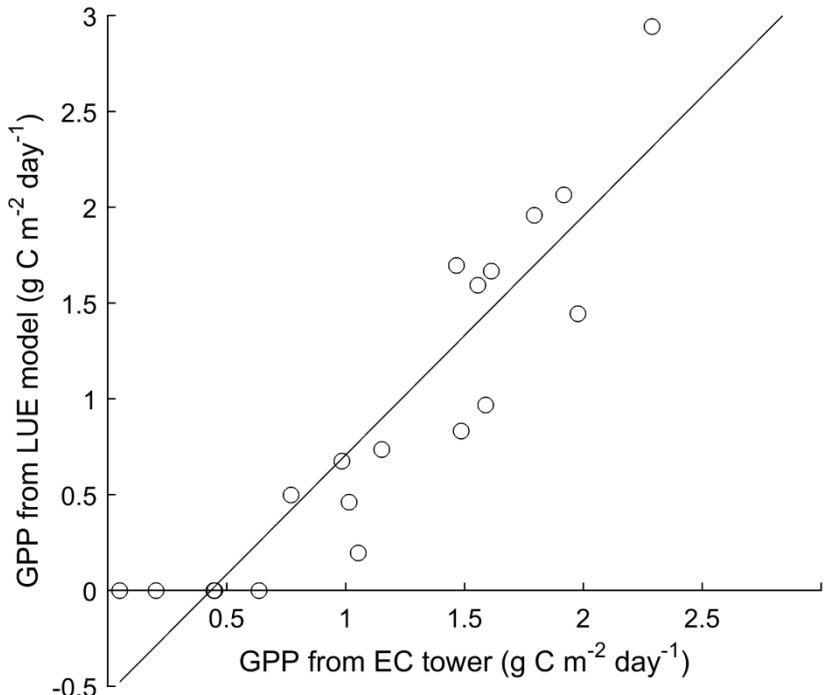

**Figure 8. Correlation between GPP from the EC tower and LUE modelled GPP for the year 2012, with insect outbreak.** **GPP$_{lue\ defoliated}$ = -0.54 $\times$ 1.25GPP$_{EC}$, $R^2$ = 0.83 and N = 19.**

### 3.3.2 Defoliated areas and quantifying the insect outbreaks impact on annual GPP

In the year 2012, with the most widespread defoliation in this study, 76% of the 100 km$^2$ forests were defoliated (Table 2 and Fig. 9). In 2004 and 2013, 53% and 55% of the forests were defoliated, respectively. The mean annual reduction in regional GPP due to the insect outbreaks for the three outbreaks studied was 15±5 Gg C yr$^{-1}$ according to Method 1, with the largest outbreak in 2012 with a negative impact on regional GPP of 18±6 Gg C yr$^{-1}$ (Table 2). The average annual regional GPP in the study area, derived with the LUE model (Eq. 12) and the five years without insect outbreak, was 41±12 Gg C yr$^{-1}$, which gives a reduction in 2012 of 44%. The impacts of the outbreaks in 2004 and 2013 were reduction in regional GPP of 32% and 34%, respectively. There were no differences in the GPP reduction per square meter between the outbreak years.

When a LUE model was applied to model GPP also during defoliation events (Method 2) the mean annual decrease in regional GPP was 15±5 Gg C yr$^{-1}$, which is the same estimate as with Method 1. The regional GPP loss in 2012 was 20±7 Gg C yr$^{-1}$, which is slightly higher compared to Method 1. In the year 2004 the two methods resulted in similar decreases in GPP, while the GPP decrease was larger with Method 1 in 2013. Differences in GPP loss per square meter between the years were larger with Method 2: 190±67 g C m$^{-2}$ yr$^{-1}$ in 2013 was the lowest GPP loss, and 270±95 g C m$^{-2}$ yr$^{-1}$ in 2012 was the largest GPP loss.

Table 2. Defoliated area (km$^2$) and annual reduction in GPP[a] (Gg C yr$^{-1}$) for the three years with insect defoliation since the year 2000. The total area with forest cover is 100 km$^2$.

| Year | | 2004 | 2012 | 2013 |
|---|---|---|---|---|
| | Defoliated area (km$^2$) | 53 | 76 | 55 |
| GPP decrease[a] | Mean (g C m$^{-2}$ yr$^{-1}$) | 240±72 | 240±72 | 240±72 |
| Method 1 | Total (Gg C yr$^{-1}$) | 13±4 | 18±5 | 14±4 |
| (GPP reduction factor) | Total (%) | 31 | 45 | 33 |
| GPP decrease[a] | Mean (g C m$^{-2}$ yr$^{-1}$) | 250±88 | 270±95 | 190±67 |
| Method 2 | Total (Gg C yr$^{-1}$) | 13±5 | 20±7 | 10±4 |
| (Defoliation LUE model) | Total (%) | 33 | 49 | 25 |

[a]GPP for undisturbed conditions is derived with the LUE model (Eq. 12) and as the mean of the five years without insect defoliation.

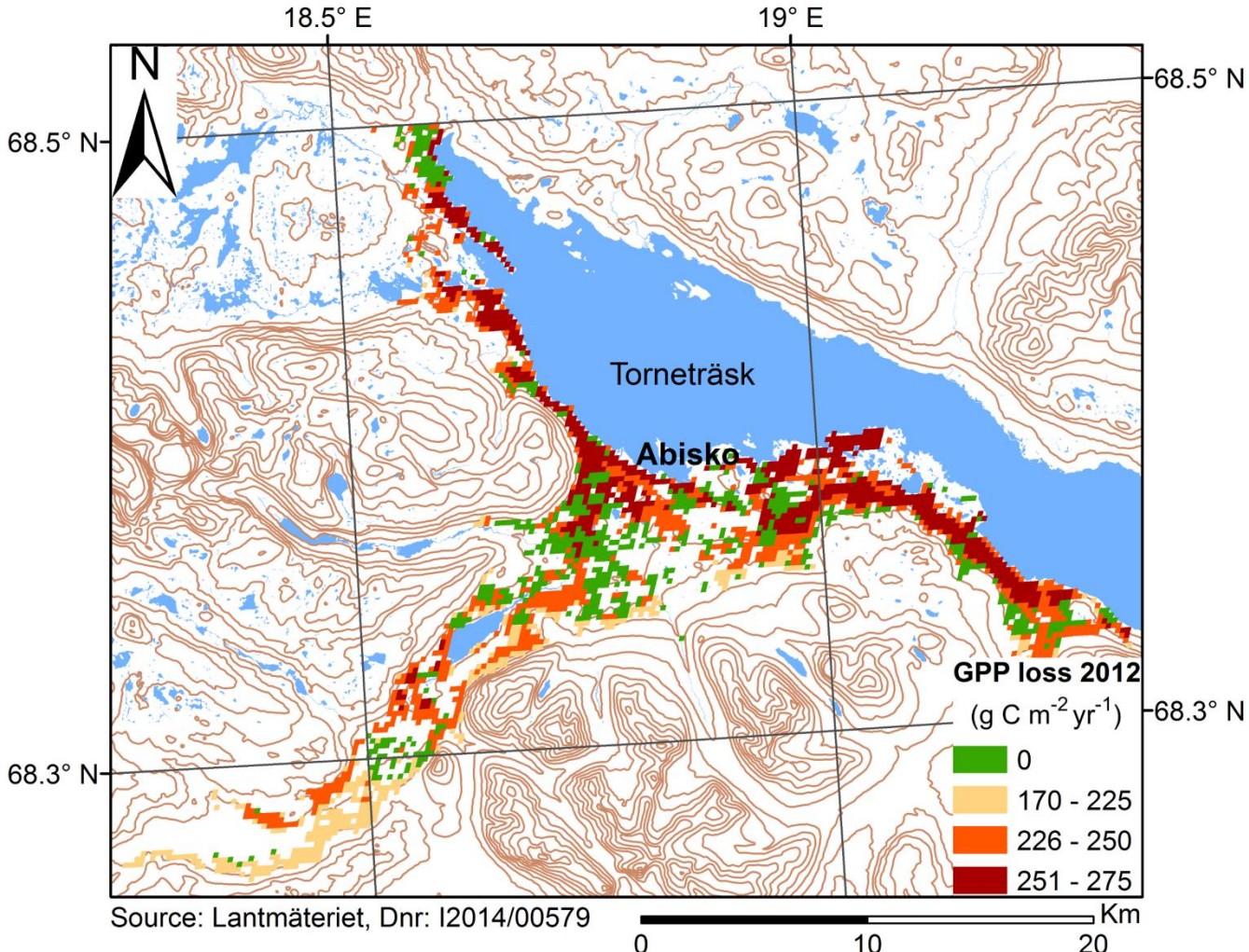

**Figure 9.** Reduction in annual GPP (g C m$^{-2}$ yr$^{-1}$) due to the outbreak of autumnal moth and winter moth in 2012 computed with a LUE model also for defoliation (Method 2). One standard deviation of the GPP losses is estimated to 35% of the given values. Areas with only the background map have a canopy cover less than 50% or are outside the study area shown in Fig. 1. The reference system is SWEREF99 TM and latitude and longitude are in WGS84. Source of background map: Lantmäteriet (Dnr: I2014/00579).

We compared the differences in GPP decrease between Method 1 (GPP reduction factor) and Method 2 (two LUE models) to study if Method 2 performed better for MODIS pixels where the birch trees recovered later in the growing season. For all years the mean differences in GPP loss (g C m$^{-2}$ yr$^{-1}$) between the methods were lower for pixels that recovered later in the growing season. These results suggest that Method 2 captured some of the refoliation, though the differences are small and within the error margin.

**Table 3. Differences in GPP loss (g C m$^{-2}$ yr$^{-1}$) between Method 1 and Method 2 for MODIS pixels with recovering foliage later in the season, and pixels with no refoliation according to the defoliation monitoring method. Higher GPP loss with Method 2 gives negative values.**

| Year | 2004 | 2012 | 2013 |
|---|---|---|---|
| Refoliated pixels | 48% | 14% | 52% |
| Difference, refoliated (g C m$^{-2}$ yr$^{-1}$) | -9±3 | -19±7 | 57±20 |
| Difference, non-refoliated (g C m$^{-2}$ yr$^{-1}$) | -15±5 | -24±8 | 54±19 |

## 4 Discussion

This study has shown a substantial setback in GPP caused by insect defoliation in a subarctic deciduous forest in northern Fennoscandia. At the EC tower, GPP decreased by 260 g C m$^{-2}$ yr$^{-1}$ (59%) during the outbreak in 2012 compared to the mean of undisturbed years. In the entire study area annual mean values of decrease in GPP ranged from 190±67 to 270±95 ±g C m$^{-2}$ yr$^{-1}$. The total decrease in regional GPP, due to the three insect defoliation events studied here was estimated to be 45±14 Gg C, which is of the same magnitude as the average annual regional GPP of 41±12 Gg C yr$^{-1}$ for single years with no

disturbances. During the most severe outbreak year (2012), the annual regional GPP loss was nearly 50% (20 Gg C yr$^{-1}$), with 76% of the 100 km$^2$ birch forests in the study area defoliated. In this study we have estimated the impact on GPP only but we noted that during the outbreak in 2012 the decrease in R$_{eco}$ was larger than the decrease in GPP during the growing season around the EC tower. Respiration is affected by insect outbreak in two ways: (1) Autotrophic respiration is reduced as defoliated trees cannot photosynthesize, and (2) heterotrophic respiration increases when dead larvae decompose. The

amount of carbon respired by larvae is likely to be the same as the amount of carbon in eaten leaves, so we should only observe a shift of respiration in time. In addition, larvae transport nutrients from trees to fungi and bacteria living in soil, which further increase respiration. The increase in heterotrophic respiration did not offset decrease in autotrophic respiration, and R$_{eco}$ for outbreak year was decreased in comparison to non-disturbed years. This study also highlights the advantage of combining EC data and remote sensing data, by applying data from an EC tower to calibrate an LUE model, and applying

satellite data to estimate the impact on GPP over larger areas. EC measurement alone cannot be extrapolated with high accuracy if the spatial and temporal extent of an outbreak is unknown, and the LUE model could not be developed without EC data. The combination facilitates wall-to-wall mapping of forest disturbances, and quantitative estimates of the impacts on primary productivity.

There are, however, limitations in the study that must be considered. One major challenge is to establish baseline conditions

for GPP in areas with reoccurring insect outbreaks, as in Abisko. As a comparison, Olsson et al. (2016a) tested a defoliation detection method on the outbreak in Abisko in 2013 and achieved the highest detection accuracies when the baseline conditions were based on the six years with highest NDVI values in the period 2000–2012. In this study the annual GPP for years without disturbances was estimated as the mean of the five years without insect outbreak and with available EC data. It

is likely that some of these five years were influenced by the insect outbreak in 2012. The two years prior to when the insect populations reached outbreak levels (2010 and 2011) had lower annual GPP than the years 2007 and 2009 (Table 1), and it is likely that GPP in 2014 was influenced by the insect defoliation in 2012 and 2013. Heliasz (2012) suggested that GPP reaches pre-outbreak levels 2–3 years after an outbreak, and Hoogesteger & Karlsson (1992) showed that LAI returned to

pre-defoliation levels two years after 100% artificial defoliation even though tree ring width was lower than normal at least three years after the experiment. For the birch forests to fully recover from severe outbreaks may take decades (Tenow & Bylund 2000). To get an indication of the potential influence on GPP by insect defoliation for the non-outbreak years we modelled GPP based on PAR for the years with data available from the EC tower and compared with EC derived GPP (see supplementary materials). The result showed that measured GPP at the EC tower, and GPP modelled with PAR data, were

similar in 2007 and 2009. In the two years prior to the outbreak (2010 and 2011), measured GPP was lower than PAR modelled GPP, indicating that there were signs of defoliation by growing larval population. Also in 2014 when the birch forests were recovering, measured GPP was lower than PAR modelled GPP. During the insect outbreak in 2012 measured annual GPP was 290 g C m$^{-2}$ yr$^{-1}$ lower than PAR modelled GPP, which is larger than the decrease of 260 g C m$^{-2}$ yr$^{-1}$ applied in this study. In addition, we ran the LUE model with meteorological data from ANS for the year 2008 to fill the gap

in the time-series with measured GPP and to study how well it agreed with the years 2007 and 2009. According to the LUE model annual GPP at the EC tower was 440 g C m$^{-2}$ yr$^{-1}$ in 2008, which agrees with the GPP value for undisturbed years of 440 g C m$^{-2}$ yr$^{-1}$ that we are applying in the study. However, since years that are influenced by pre-outbreak defoliation as well as a recovery year are included as undisturbed years it is likely that the baseline GPP applied in this study is lower than GPP for undisturbed conditions. This is also indicated by the larger difference between PAR modelled and measured GPP in

2012 and suggests that the estimated decreases in GPP due to insect outbreaks in this study are on the lower side.

Another limitation is the assumption that no other factors than insect outbreaks influence annual GPP, even though it is likely that also meteorological conditions influence GPP. The comparison between EC derived GPP and PAR modelled GPP suggests that only two years with EC data represent undisturbed forest; hence, the amount of data from the EC tower is too small to study correlations between EC derived GPP and meteorological variables. Instead we studied correlations between

NDVI and meteorological data from ANS, where we used the mean of the highest $NDVI_{DL}$ value of each year derived from 200 MODIS pixels with birch forest. To minimize the influence of insect induced defoliation we excluded the outbreak years and years prior to and after outbreaks. No linear correlations between PAR and GPP were found. There were, however, negative correlations between temperature and seasonal maximums of $NDVI_{DL}$, with the strongest correlation between NDVI and the mean temperature in May–June. The influence of temperature on NDVI was weak, and due to the estimated

uncertainties of the LUE model of 30% we did not include these correlations in the analysis. However, with data from the EC tower available for more years it would be a potentially important improvement to include meteorological data when estimating the decrease in annual GPP.

There are also uncertainties in the LUE model. The relationship between $fAPAR_{8day}$ and $NDVI_{DL}$ (Eq. 11) was estimated from two growing seasons without disturbances. Due to larvae disrupting the PAR-sensors there were no fAPAR data available

from the outbreak years, hence, Eq. 11 was used also for defoliation events. Furthermore, the relationship was derived from fAPAR obtained from the upper canopy, which may not be representative for the entire forest, since the relationship between $fAPAR_{8day}$ and $NDVI_{DL}$ is likely to vary with understory and forest densities in the study area. The relationship is also likely to vary with varying understory responses due to defoliation, which may influence the estimated decreases in annual GPP.

Accounting for these uncertainties would require more data, both for the fAPAR and NDVI relationship, and detailed land cover data, which would make the model more complex. Hence, we assume this limitation to be acceptable, and since the aim of the study was to estimate the influence of defoliation of the birch trees, we considered $fAPAR_{canopy}$ to be the most suitable variable. Another potential limitation is that the LUE model developed for years with defoliation seems to underestimate GPP for values lower than about 1.5 g C m$^{-2}$ day$^{-1}$ (Figure 8). However, for the outbreak year with available

EC data (2012) the underestimated values from the LUE model are mainly due to a cold spring that resulted in a large reduction factor ($f_{8day}$). During the main growing season LUE modelled and EC derived GPP agrees well, which increases confidence in the modelling.

It may also seem surprising that the difference in $NDVI_{DL}$ was comparably low in relation to the difference in light use efficiency. It is, however, known that NDVI saturates for high LAI and that small changes in NDVI can be associated with

large changes in LAI (e.g. Myneni et al. 2002). The light use efficiency on the other hand can decrease substantially with lower LAI since more leaves will operate in the light-saturated portion of the photosynthesis (e.g. Medlyn 1998). There are also uncertainties in how well the EC tower footprint represents the entire study area. Heliasz (2012) utilized a permanent EC tower as reference and a mobile EC tower to study variability in carbon exchange in the birch forests around Abisko and concluded that there were only minor differences in GPP at seven sites during the peak growing season in 2008 and 2009.

Hence, we consider the EC tower footprint to be representative for the study area.

The accuracy of the defoliation detection method also influences the results of the study. The method missed 26% of the defoliated MODIS pixels and misclassified 39% of the undisturbed pixels as defoliated in the evaluation data used by Olsson et al. (2016b). This implies that the defoliated areas in 2004 and 2013 were slightly overestimated, while the defoliated area in 2012 is likely underestimated, though the impact on the total numbers is likely small. It should also be considered that

20% of the forests in the study area were excluded since they are located in MODIS pixels with < 50% forests cover. Thus, the current estimate of a total reduction in GPP may be conservative.

A limitation with the developed LUE model for large-area estimates is that it includes observed meteorological data (temperature and PAR). An alternative for running the model over larger areas would be to use modelled meteorological data (Olofsson et al. 2007; Schubert et al. 2010). There are also uncertainties related to the temperature data utilized. The gradient

applied to model mean temperatures depending on altitude is likely to give accurate estimates in the study area. However, minimum temperatures are more uncertain since cold air can drain downhill and accumulate in valleys and low areas, rather than decrease with altitude. Altogether, since the EC tower is located on a small ridge in the lower, flat parts of the study area, we anticipate that the temperatures there are not substantially lower than the area in general. We compared with lowest daily temperature from Abisko research station, which is located near the spectral tower 10 km to the west (Fig. 1), and at a

slightly higher altitude than the EC tower. For all periods with frost events during the early season, i.e. when the lowest temperature influences $f_{MOD8}$, the mean value of absolute differences, with the coldest temperatures at the research station, was only 0.4°C. With these small temperature differences and since frost events only influence GPP in the early growing season, the impact on annual GPP was considered minor.

The defoliation detection methods used in this study gives a time-series of smoothed NDVI that captures the timing of the defoliation event as well as the potential refoliation. The LUE model on the other hand utilizes NDVI smoothed with double logistic functions. These functions do not capture the typical seasonal trajectory for years with refoliation. This is illustrated in Figure 3 where raw NDVI stays around 0.6 during the entire growing season in 2012 when there was no refoliation around the EC tower. In 2013, when there was substantial refoliation around the EC tower, raw NDVI stays around 0.6 during June,

but increase to pre-outbreak levels in early July when refoliation occurs. In 2004 the raw NDVI values has a similar pattern as in 2013 with low values (around 0.6) until early August when refoliation results in a later season peak in NDVI. This seasonal development of raw NDVI agrees well with GPP for the limited period with available EC data in the outbreak year 2004. $NDVI_{DL}$ does not capture this trajectory with sharply increasing NDVI values that levels off and starts increasing again later in the season. However, even though the actual timing of the defoliation is not captured during years with refoliation the

total growing season GPP is well modelled. A new version of TIMESAT, currently developed and tested, will capture also more detailed seasonal trajectories with smooth fitting of curves. These new curve fitting methods have a potential to improve the performance of the LUE model.

We applied two methods to quantify the impacts on GPP to study which methods performed better for refoliating birch forests. The assumption was that Method 2 would be more adaptive and adjust for differences in defoliation intensities

between MODIS pixels. Since the level of defoliation, as well as understorey responses to the defoliation, are likely to influence $NDVI_{DL}$, which in turn will influence $fAPAR$, it was anticipated that a method based on a LUE model to derive GPP during defoliation events would capture variability in defoliation levels and understory responses between MODIS pixels. Method 1, on the other hand, with a common reduction factor, does not account for local differences between pixels and is similar to upscaling the local conditions at the EC tower, even though the method has the advantage that annual GPP

for each pixel is derived with a LUE model, and hence should be more accurate than assuming that GPP for all MODIS pixels is identical to GPP at the EC tower. For the years 2004 and 2012, the two methods resulted in similar estimates of the GPP loss with slightly larger decrease in GPP for Method 2. In 2013, the difference between the methods was larger with the highest decrease in annual GPP for method 1. One possible explanation for the smaller decrease in annual GPP according to Method 2 for the year 2013 is that the growing season seems to have been shorter and that refoliation started earlier and was

stronger in 2013 compared to 2004; this is indicated by the seasonal developments of NDVI. It should also be noted that higher NDVI might be due to increasing growth of understory grasses favoured by the changed light conditions due to defoliation (Karlsen et al. 2013) rather than recovering birch.

The impact of insect outbreaks on the carbon balance has been quantified in earlier studies: Heliasz et al. (2011) studied the impact on NEE of the autumnal moth and winter moth outbreak in Abisko in 2004, but these measurements started on July 2,

which was around 10 days after the larvae reached peak densities, which most likely resulted in an underestimated reduction in NEE. To facilitate a comparison between the outbreak years 2004 and 2012, we computed GPP for the period July 2 to September 30 for all years with EC data. This indicated that the two outbreak years had similar impact on the carbon balance during the period studied with a GPP loss of 210 g C m$^{-2}$ yr$^{-1}$ in 2004 and 200 g C m$^{-2}$ yr$^{-1}$ in 2012 compared to years without

disturbance. Furthermore, the loss of 200 g C m$^{-2}$ yr$^{-1}$ in the year 2012 and for the same time period as studied in the year 2004, compared to the GPP loss of 260 g C m$^{-2}$ yr$^{-1}$ for the entire growing season in 2012, suggests that the impact on NEE was underestimated by Heliasz et al. (2011). Clark et al. (2010) found the highest difference in NEE between undisturbed years and years with severe defoliation by the gypsy moth in New Jersey, USA, to be 266–480 g C m$^{-2}$ yr$^{-1}$, and Clark et al (2014) found that mid-day NEE during complete defoliation was 14% of pre-defoliation rates. Allard et al. (2008) noted that

cumulative NEE was lower during a year with insect defoliation compared to years without disturbances; however, the low NEE value might to a large extent have been caused by a dry spring. Brown et al. (2010) found that a mountain pine beetle outbreak turned a forest into a carbon source; no pre-outbreak EC data were available to quantify the impact on NEP, but recovery after the outbreak was faster than anticipated (Brown et al. 2012). It should be noted that the mountain pine beetle feed within the phloem and directly kills trees, while the moth species discussed above are defoliators that usually only kill

trees in cases of severe and repeated outbreaks (Hicke et al. 2012). Modelling studies have also found that forests have changed from sinks into sources of carbon, in some cases for extended periods (Kurz et al. 2008a; Dymond et al. 2010; Schäfer et al. 2010; Medvigy et al. 2012). However, to our knowledge, this is the first study that has utilized remote sensing data and developed a LUE model calibrated with EC data, to both quantify and map the spatial extent of the impact of defoliating insects' outbreaks on GPP.

The results of this study could help to reduce uncertainties in the impact of insect outbreaks on primary productivity as well as to improve carbon budgets by including insect induced defoliation. For the mountain birch forests in this study the estimated reduction in annual GPP, compared to years without disturbances, was 50% when there was limited refoliation in the study area. For years with widespread refoliation, the annual GPP losses were about one third of GPP for years without disturbances. In addition, the spatial and temporal mapping of insect defoliation provided by remote sensing is important for

accurate simulation of the carbon dynamics. Furthermore, the outbreak area included in this study is only a fraction of the 10,000 km$^2$ estimated to having been severely defoliated in northern Fennoscandia during the period 2000–2008 (Jepsen et al. 2009). Assuming that the conditions were similar over northern Fennoscandia, the insect defoliation over these vast areas would result in a potential total regional GPP loss for the time period of the magnitude 2–3 Tg C. Models not accounting for such recurring disturbance events would seriously overestimate the ability of these forests to absorb atmospheric $CO_2$.

**5 Conclusions**

This study showed, with the aid of MODIS NDVI and eddy covariance data, a substantial loss in regional GPP due to insect induced defoliation in subarctic deciduous forests in northern Fennoscandia. The estimated mean annual decrease in regional

GPP for a year with insect outbreak was 15±5 Gg C yr$^{-1}$± in the study area of 100 km$^2$. This should be compared with the average annual GPP of 41±12 Gg C yr$^{-1}$ for years without disturbances. In the most severe outbreak year (2012) 76% of the birch forests were defoliated and annual GPP was merely 50% of GPP for years without disturbances.

The study also demonstrated the use of remote sensing data to both monitor the spatial extent of the defoliation and to estimate the impact on the primary productivity of these defoliation events. The insect disturbance is shown to have major impacts on the primary production of the sub-arctic forest; consequently, the derived methods, based on combining remote sensing and eddy covariance measurements, are of major importance to support carbon balance estimates over large areas.

The authors declare that they have no conflict of interest.

**Acknowledgement**

We want to thank Fredrik Lagergren, Department of Physical Geography and Ecosystem Science, Lund University, for valuable discussions about the LUE model, and Annika Kristoffersson at Abisko Scientific Research Station for providing us with meteorological data. The Land Processes Distributed Active Archive Center (LP DAAC) is acknowledged for providing access to the MODIS data (https://lpdaac.usgs.gov/data_access/data_pool). The project was mainly funded by a research grant from the Swedish National Space Board to Lars Eklundh.

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
