# Peer review of "Mapping the reduction in gross primary productivity in subarctic birch forests due to insect outbreaks"

_Biogeosciences, 2016_

## Referee Comment (RC1) · Anonymous Referee #1 · 19 Oct 2016

In "Mapping the reduction in carbon uptake in subarctic birch forests due to insect outbreaks", the authors combined data from an eddy covariance (EC) tower, a spectral tower, and the MODIS-based Normalized Difference Vegetation Index (NDVI) to develop light use efficiency (LUE) models of gross primary productivity as a function of NDVI in northern Sweden forests, for years without or with defoliation caused by two moth species. Using MODIS NDVI to identify defoliated pixels over the study area, they then compared the regional GPP for 5 no-outbreak years (based on the no-outbreak "GPP_lue" model) and 3 outbreak years (based on a GPP reduction factor derived from EC tower data (Method 1) or the outbreak "GPP_lue,defoliated" model (Method 2)). The manuscript is concise, presents many informative Figures, and addresses some uncertainties through Monte Carlo analysis. Overall, I think the manuscript fits within the scope of Biogeosciences. However, the use of the mean GPP over no-outbreak

years as the basis to estimate GPP reduction during outbreaks is questionable and the manuscript is not sufficiently well written to warrant publication in its present form. In addition to addressing the issues listed below, the authors should go through the manuscript to correct the many typos I did not highlight.

MAJOR COMMENTS

1. Using the mean GPP over no-outbreak years as the basis to estimate GPP reduction caused by outbreaks seems inadequate for two different reasons. First, differences in weather may affect both GPP and outbreak occurrence/severity. If the two variables are indeed correlated across years, the approach likely causes a bias. This should be checked; no need for fancy statistical tests, just compare, based on the GPP_lue model, the mean GPP and its standard deviation for outbreak vs. no-outbreak years over pixels that have never been defoliated. Second, GPP in pixels previously defoliated is unlikely equal to what it would have been if no outbreak had occurred. For example, canopy trees might have not fully recovered yet (hence underestimating the no-outbreak GPP) or total tree+understory productivity might increase for a few years due to the defoliation-caused growth release of the understory (hence overestimating the no-outbreak GPP). The authors should rather have estimated the no-outbreak GPP in pixels that have been defoliated up to X years before (with X to be defined; maybe 3-5 years?) based on the NDVI_DL values of neighbouring pixels that have never been defoliated. (More precisely: for each defoliated pixel, define a window large enough to include never-defoliated pixels, but small enough to have similar conditions. Then, compute the mean NDVI_DL difference between the defoliated pixel and the never-defoliated pixels over all years prior to the (first) defoliation event in the defoliated pixel; let's say NDVI_DL was on average 10% higher in the defoliated pixel. Finally, for the X years after the defoliation event (excluding the defoliation year itself), estimate the annual no-outbreak GPP in the defoliated pixel with the GPP_lue model, but with a value of NDVI_DL 10% higher than the mean annual NDVI_DL value over the neighbouring never-defoliated pixels [instead of using the annual MODIS NDVI_DL value in the defoliated pixel].) Ideally, the authors should re-do their analyses using this new approach. At a minimum, the authors must use this new approach for >100 randomly-selected defoliated pixels and see to which extent it affects their results.

2. In the Discussion, the authors must at least explicitly acknowledge four major methodological weaknesses; when possible, explain the likely impact of each weakness (i.e., under- or overestimating defoliation-caused GPP losses) and propose a way to address the weakness. (1) fAPAR was based on measurements for the upper canopy only, so it is unclear to which extent the fAPAR vs. NDVI_DL relationship applies to the entire forest. This is particularly critical due to the (possible) growth release of the understory highlighted by the authors. (2) The fAPAR vs. NDVI_DL relationship was derived for undefoliated years (2010-2011) only, yet was also used in the GPP_lue,defoliated model. Why not developing a fAPAR vs. NDVI_DL for defoliated conditions (no defoliation event at the spectral tower over the entire study)? (3) The defoliation detection algorithm missed 26% of defoliated areas and misclassified 39% of undefoliated areas. (4) How representative was the EC tower footprint of the entire study area, both during outbreak and no-outbreak years? This is critical, as EC tower data provided the basis for all GPP estimates through the values of \epsilon_max, \epsilon_max,def, and the GPP reduction factor.

3. I object to providing the 3-year *total* GPP reduction caused by defoliation, as this inadequately inflate numbers. Please provide the 3-year *mean* reduction instead throughout the text, making it clear the reduction is for outbreak years only (not the mean values over all years since 2000): Abstract; P16, L15 to P17, L1; P17, L7-8; P19, L5-7; P19, L15; P21, L18-20.

4. P3, L15-26. I have various issues with the text from "Since near-linear" to "(Liljedahl et al. 2011)". First, it should be in the Methods; I suggest merging at the beginning of Section 2.3. Second, units are not provided for the variables and the equation is not numbered; please number *all* equations and provide units for *all* variables throughout the text (even when unitless; e.g., NDVI). Third, the sentence starting on L20 is

cumbersome; if kept in Section 2.3, I suggest re-writing along the following lines: "The light use efficiency coefficient varies between vegetation types and the influence of meteorological conditions is accounted for through reduction factors for temperature and vapour pressure deficit [...]". Fourth, the sentence starting on L23 is an overstatement because: 1) temperature is not always the main limiting factor in cold climates (Nemani et al. 2003; Beer et al. 2010); and 2) neither Bergh et al. (1998) nor Lagergren et al. (2005) really tested for the impact of factors other than temperature in their studies that covered only two sites in Sweden. The authors should acknowledge that they assumed accounting for temperature only was sufficient in their study region, an assumption supported (but not demonstrated) by Bergh et al. (1998) and Lagergren et al. (2005). Fifth, delete the sentence starting on L25: water stress is a major limiting factor in some boreal and other forests, not just for "ecosystems dominated by non-vascular plants".

5. P6, L12. Why wasn't fAPAR_canopy also smoothed with TIMESAT before the regression?

6. P7, Figure 3. The TIMESAT smoothing removed a second NDVI 'peak' each year. Please explain what was the origin of this (wintertime?) second annual peak and why removing it was OK.

7. P12, L4-6. I do not understand why Method 1 should not also capture the effect of refoliation: the EC tower data should account for the post-refoliation increase in GPP, no? Unless no refoliation occurred within the EC tower footprint after the 2012 outbreak?

8. P14, Figure 6. Many readers will likely expect defoliation to substantially decrease NDVI (due to a much lower leaf area index (LAI)) and leave LUE barely affected, so they will question the defoliation results (small NDVI decrease, large LUE decrease). It would thus be helpful to explain that small reductions in NDVI are associated with large reductions in LAI (e.g., Wulder et al. 1998), while LUE can substantially decrease

for lower LAI because more leaves operate in the light-saturated portion of the photosynthesis curve (e.g., Medlyn 1998). Also, please indicate the weeks during which defoliation occurred.

9. P15, Figure 7 and P16, Figure 8. Give the equations for the regression lines on Figures 7 and 8. The line seems pretty close to 1:1 with zero intercept in Figure 7 (hence no bias for no-outbreak years), but the GPP_lue,defoliation model in Figure 8 seems to underestimate GPP over most of the range of actual values (i.e., up to $\sim2$ gC m^-2 day^-1); this should be added to the list of weaknesses in the Discussion (see comment #2).

10. P18, L7-10. Unless I am mistaken, the authors do not correctly interpret these results. If Method 2 captured the effect of refoliation whereas Method 1 did not, then Table 3 results should have a higher absolute value for refoliated than non-refoliated pixels *regardless* of the actual sign. This is what was obtained, suggesting that Method 2 did better capture refoliation's effect; however, the differences between refoliated and non-refoliated pixels are always within the error margin, so this better performance is marginal. (The sign of Table 3 values (negative for 2004 and 2012, positive for 2013) is not directly related to the effect of refoliation, but to the bias between the two methods.)

11. P19, L26 to P20, L3. I do not think these explanations for Table 3 results (i.e, the difference in GPP loss between Method 1 and Method 2) are appropriate. The refoliation argument does not hold because, as noted by the authors, both 2004 and 2013 had high refoliation yet had opposite signs for Method 1 minus Method 2. Furthermore, the lower GPP losses in 2013 for Method 2 were basically the same for non-refoliated and refoliated pixels (Table 3). The argument of "uncertainties in \epsilon_max,def" does not seem appropriate either, because the same value was used for all years (so why a sign change in 2013 for Method 1 minus Method 2?). The argument of higher NDVI due to understory growth would work if this growth was stronger in 2013 compared to 2004 and 2012; are there reasons to believe this was the case? Here is another hypothesis that could account for the much lower mean GPP reduction in 2013 under Method 2 compared to all other values (and hence account for the sign change in 2013): could it be that growing conditions were better in 2013, thereby leading to higher f_8day and/or PAR_8day used in GPP_lue,defoliated compared to the mean f_8day/PAR_8day no-outbreak values used in GPP_lue?

MINOR COMMENTS

12. Throughout the text: when possible replace the vague "carbon uptake" expression by the more accurate term applicable (GPP, NEE, etc.). Therefore, the title should be: "Mapping the reduction in gross primary productivity due to insect defoliation in subarctic birch forests". In the Discussion, the authors should stress that their results are not for the net carbon balance, but for GPP only; based on Heliasz et al. (2011), would it be possible to speculate whether, on a percentage basis, NEP losses should be higher or lower than GPP losses?

13. Throughout the text: please use the "regional GPP" expression when applicable, to make it clearer the values are for the total GPP over the study region.

14. P1, L18-20. I have various issues with this sentence; I suggest re-writing along the following lines: "In the study area of 100 km^2, the results suggested a mean regional GPP decrease of XX +/- YY Gg C yr^-1 for the three outbreak years (2004, 2012, and 2013), compared to a mean regional GPP of 41.1 +/- 12 Gg C yr^-1 for the five years without defoliation".

15. P2, L1. Reference(s) should support the "a warmer climate is likely to increase forest productivity" part of the sentence. Here are some suggestions: Pastor and Post (1988), Nemani et al. (2003), or Boisvenue and Running (2006).

16. P2, L21. Since Brown et al. (2012) is already cited later on, I suggest mentioning it here too.

17. P3, L3. Please give the temporal coverage of Landsat.

18. P3, L13. It is Monteith and Moss (1977).

19. P3, L27-28. Bright et al. (2013) already used remote sensing (Landsat to identify trees killed by the mountain pine beetle) and a LUE model (MODIS GPP results, which are based on LUE) to quantify the impact of an insect outbreak on carbon uptake. To my knowledge, the authors can still claim being the first ones to do it for defoliators.

20. P3, L30-34. Delete the "This combination of [...]" sentence or merge it with the previous one (it is repetitive). Delete the "The method was developed [...]" sentence (it is repetitive with "Our main study objective [...]").

21. P3, L34 to P4, L2. Delete or merge with the Methods.

22. P4, L2-4. The sentence is confusing, as it seems to imply that results will be provided for every year "between 2000 and 2015". I suggest re-writing along the following lines: "Our main study objective was to compare GPP for years with (2004, 2012, and 2013) and without (2007, 2009, 2010, 2011, and 2014) insect outbreak in the birch forest of a subarctic valley of northern Sweden".

23. P5, L18 to P6, L1. The purpose of "quality classes based on QA data" is not clear: how where these classes used in the analysis?

24. P6, L4. Replace "fraction canopy absorbed PAR" by "fraction of absorbed PAR by the canopy". Similar comment for "canopy absorbed PAR" in Figure 4 caption.

25. P6, L8. Is the "pure canopy absorbed PAR" different from fAPAR_canopy? If yes, quickly explain what is the difference and why it matters. If not, replace the expression by "fAPAR_canopy".

26. P6, L11. I would provide here (after "NDVI_DL") — instead of on P3, L20 — the reference to Myneni & Williams (1994) that supports the linear relationship.

27. P7. I suggest combining Figure 2 with Figure 1. For all multi-panel Figures, please add letters to each panel and refer to the appropriate panel in the text (e.g., Fig. 5a).

28. P8, L28. Please put here (instead of on P11, L8-10) the sentence about the value

of the temperature lapse rate.

29. P9, L2. The units for PAR_8day should be MJ mˆ-2 day-1.

30. P9, L5. Add something like "(see Section 2.3.3)" so that readers know the value of \epsilon_max will be discussed later. Similar comments for GDD_thresh (P9, L16) and T_thresh (P10, L2).

31. P9, Equation (7). Replace the current condition on the middle line by "-8 $\leq$ T_min8 < -3".

32. P10, L3. T_mean8 was never negative for such a northern site at the end of September? If negative values occurred, explain how they were dealt with (Equation (8) suggests a negative value for f_8day, which would make no sense).

33. P10, L6. Here and in Figure 5, replace "RMS" by "RMSE".

34. P10, Equation (9). Replace "\epsilon" by "\epsilon_max".

35. P11, L8. Delete the first sentence (repetitive with the previous paragraph).

36. P11, L15-16. The text should be expanded, because at this point the reader is still unaware that an outbreak occurred in 2012 within the EC tower footprint: please state this explicitly here and add what was the percentage of defoliation within the tower footprint.

37. P12, L1-2. The sentence is cumbersome; I suggest re-writing along the following lines: "For each year with insect outbreak, the regional reduction in GPP was computed by summing, over all pixels identified as defoliated, the difference between the mean GPP for no-outbreak years and the GPP for this specific outbreak year".

38. P13, L6. For consistency, replace "GDD threshold" by "GDD_thresh", and "temperature factor" by "T_thresh".

39. P14, Figure 6 legend. For consistency, replace "f_MOD8" by "f_8day".

40. P15, L8. Here and throughout the text (including Figures): provide full units for local, mean, and regional GPP (i.e., with yrˆ-1) even when writing "annual".

41. P16, L14-15. This sentence is repetitive with P11, L4-6.

42. P17, L1-2. Make it clearer that the 41.1 value was computed as the mean over no-outbreak years, based on the GPP_lue equation (add a number) given on P14. Please also add this information as a footnote to Table 2.

43. P18, Figure 9 caption. Replace "birch moth outbreaks" by "outbreaks of autumnal moth and winter moth".

44. P19, L2. The word "demonstrated" seems too strong; similar comment for P21, L17.

45. P19, L3. Replace "decreased with 260 g C mˆ-2" by "decreased by 261 g C mˆ-2 yrˆ-1" (based on Table 1, the difference is 261). Similar comment for P20, L25.

46. P19, L5. According to Table 2, the highest value is 265 +/- 93 (not 244 +/- 73).

47. P20, L31. Replace "turned a forest into a carbon sink" by "turned a forest into a carbon source". Stands were carbon sinks only during the growing season; the main point here is that insect-caused mortality led to negative annual NEP... although these stands seem to be recovering quickly, as correctly noted by the authors in the second part of the sentence!

48. P21, L1-2. For (partially) contrasting modeling results about the effect of insect outbreaks on the carbon balance, see Seidl et al. (2008), Albani et al. (2010), and Landry et al. (2016) (the last one is not the same reference as already cited in the manuscript).

49. P21, L10-11. Delete the whole "since it has been suggested [...] (Medvigy et al. 2012)" part of the sentence. The "spatial distribution of defoliation" issue addressed by Medvigy et al. (2012) dealt with tree-level defoliation within a stand (e.g., 100%

defoliation of 40% of trees vs. 40% defoliation of 100% of trees). This is not something addressed here, nor is MODIS-like remote sensing appropriate for this.

50. P21, L13-14. Delete the sentence: there is no firm basis for such an extrapolation, and the number would then likely end up being cited by future studies...

REFERENCES

Albani et al. (2010). Canadian Journal of Forest Research 40, 119-133

Beer et al. (2010). Science 329, 834-838

Bergh et al. (1998). Forest Ecology and Management 110, 127-139

Boisvenue and Running (2006). Global Change Biology 12, 862-882

Bright et al. (2013). Journal of Geophysical Research: Biogeosciences 118, 974-982

Brown et al. (2012). Agricultural and Forest Meteorology 153, 82-93

Heliasz et al. (2011). Geophysical Research Letters 38, L01704

Lagergren et al. (2005). Plant, Cell & Environment 28, 412-423

Landry et al. (2016). Biogeosciences 13, 5277-5295

Medlyn (1998). Tree Physiology 18, 167-176

Medvigy et al. (2012). Environmental Research Letters 7, 045703

Myneni & Williams (1994). Remote Sensing of Environment 49, 200-211

Nemani et al. (2003). Science 300, 1560-1563

Pastor and Post (1988). Nature 334, 55-58

Seidl et al. (2008). Forest Ecology and Management 256, 209-220

Wulder et al. (1998). Remote Sensing of Environment 64, 64-76

---

## Referee Comment (RC2) · Anonymous Referee #2 · 1 Nov 2016

The authors present an interesting article combining remote sensing data with eddy covariance measurements and a simple model to estimate changes in the carbon exchange of a birch forest due to moth outbreaks. The article is generally well written (particularly at the beginning), but the treatment of results needs more detail. There are a few methodological issues that need to be resolved or, at the very least, acknowledged and discussed in detail. Providing the limitations of the approach are properly considered and addressed, I believe this article should be suitable for publication.

Specific comments:

In several places the approach seems to take too simplistic a view, without properly discussing the assumptions or their impact. For example, a key result presented is the difference between GPP derived from EC measurements in an outbreak year (2012)

compared to five other years without insect outbreaks. Unfortunately, it seems EC data were only available for one outbreak year, but there is no analysis of differences between years due to factors other than insect damage. Inter-annual variability in meteorological conditions (rainfall/soil moisture, solar radiation and temperature in particular) can result in different annual total GPP. The values given in Table 1 should therefore be analysed with respect to meteorological conditions. This should also allow the 2012 value to be given some context – if the insect outbreak had not occurred, would the 2012 total be lower/higher/similar to the average based on meteorological conditions alone? Furthermore, is it possible that the outbreaks in 2012 and 2013 contributed to the lower GPP obtained in 2014, or is this attributable to meteorological conditions?

The comparison with the 2004 results of Heliasz et al. (2011) on page 20 is useful and indicates that closer analysis of the temporal evolution of the EC data may be beneficial. Currently data are separated into years with and without insect outbreaks. During those years with outbreaks, does the reduction in GPP over the course of the growing season agree with the timing of insect population growth/insect damage? Is this also supported by the NDVI data? In the years classified as being without insect outbreaks, are there any effects of (albeit smaller) insect populations on the GPP or NDVI values? Perhaps such analyses could offer further insight into the refoliation effect; it is currently hard to draw meaningful conclusions on this subject from the information given in the article. Some evidence to support the assumption that NDVIDL captures refoliation would also be useful; Fig 3 is not very convincing in this respect.

P15, L6-10: Section 2.2.3 states that EC data were available from 1 May to 30 Sep, covering most of the growing season. Do the EC observations and values given in Table 1 agree with this timeframe, or is it possible GPP in Table 1 is underestimated if the growing season extended beyond these dates? This study focuses on GPP, but could the authors comment on other potential impacts of the insect outbreak on the carbon balance? For example, how might respiration rates be affected, and might this impact the partitioning into Reco and GPP?

On a similar note, many of the decisions taken in the presentation of results and development of the model rely on data collected during non-outbreak years. Is the gap-filling approach also suitable in defoliated years?

P8, L10-11: More detail is needed about the quality control. Under which 'bad' weather/measuring conditions were data removed? How much data remained after quality control and what proportion was gap-filled?

P10, L11-2: Here, it is not clear which years have been used or why. The years for which the EC data are available should be stated in Section 2.2.3. Why was 2012 the only year used to calculate $\varepsilon$max with insect defoliation? Why were data from 2008 and 2013 not included/not available?

P12, L13: It is not clear how these statistics were calculated and they don't seem to follow from Fig 4. Please provide more details/clarification

P14, Fig 6: The two green lines for NDVIDL show higher NDVI values early in the growing season for the defoliated year. Possible reasons for this should be investigated and the impact on the results commented on.

Minor comments:

P2, L22: Change 'difference' to 'differences'

P2, L26: Would be useful to give the land cover type for the southern France site

P3, L17: Change to read 'of the form'

P3, L18: Number this equation and update the others accordingly

P3, L20-1: Change to read 'and with variability in meteorology'

P4, L24-6: It is not clear that these recent outbreaks are for the study site – please state the area they apply to. Also give some information about the EC tower in that study (i.e. location, and mention the flux measurements were also for the birch forest)

P5, L10: Change 'derived' to 'derive'

P6, L16-7: Change to read 'to be about'

P6, L18: Keep the order of 'east/west' the same as previously (L17). When the wind is from the west, the footprint is located to the west of the tower

P6, L24: Change 'wind speeds' to 'stability', as wind direction and stability tend to be the major controls on EC footprints

P7, Fig 3: Change y-axis of RH plot to 'NDVIDL'

P8, L16: Change 'measured the' to 'the measured'

P9, Eq3: Units on LHS do not equate to units on RHS

P11, L22-23 Suggest choosing alternative notation for GPPreduction, as it is a ratio of GPPs, rather than GPP itself

P12, L6: Change 'accurate' to 'accurately'

P13, Fig 5: Add units for RMS

P14 L5-6 and Fig 7: The text mentions 'low GPP observations' but in Fig 7 it looks as though the modelled GPP values are lower than the EC observed values, with several zero values. Please clarify

---

## Author Comment (AC1) · 12 Jan 2017

Dear Referee one. We thank you for the constructive and detailed comments and suggestions. We agree with most of the comments and have made some further analysis to improve the study. We have made most of the required updates to the manuscript.

***Major comments:***

*1. Using the mean GPP over no-outbreak years as the basis to estimate GPP reduction caused by outbreaks seems inadequate for two different reasons. First, differences in weather may affect both GPP and outbreak occurrence/severity. If the two variables are indeed correlated across years, the approach likely causes a bias. This should be checked; no need for fancy statistical tests, just compare, based on the GPP_lue model, the mean GPP and its standard deviation for outbreak vs. no-outbreak years over pixels that have never been defoliated. Second, GPP in pixels previously defoliated is unlikely equal to what it would have been if no outbreak had occurred. For example, canopy trees might have not fully recovered yet (hence underestimating the no-outbreak GPP) or total tree+understory productivity might increase for a few years due to the defoliation-caused growth release of the understory (hence overestimating the no-outbreak GPP). The authors should rather have estimated the no-outbreak GPP in pixels that have been defoliated up to X years before (with X to be defined; maybe 3-5 years?) based on the NDVI_DL values of neighbouring pixels that have never been defoliated. (More precisely: for each defoliated pixel, define a window large enough to include never-defoliated pixels, but small enough to have similar conditions. Then, compute the mean NDVI_DL difference between the defoliated pixel and the never defoliated pixels over all years prior to the (first) defoliation event in the defoliated pixel; let's say NDVI_DL was on average 10% higher in the defoliated pixel. Finally, for the X years after the defoliation event (excluding the defoliation year itself), estimate the annual no-outbreak GPP in the defoliated pixel with the GPP_lue model, but with a value of NDVI_DL 10% higher than the mean annual NDVI_DL value over the neighbouring never-defoliated pixels [instead of using the annual MODIS NDVI_DL value in the defoliated pixel].) Ideally, the authors should re-do their analyses using this new approach. At a minimum, the authors must use this new approach for >100 randomly-selected defoliated pixels and see to which extent it affects their results.*

Response: We agree that these are import points. We first respond to the issue with recovery after an outbreak and thank you for the good suggestion about how to handle this limitation. We did explore the possibility to perform the suggested analysis but decided not to go ahead with the analysis. The main motivation is the difficulty to find pixels without any defoliation. Even though there are pixels that are not detected as defoliated during outbreak years we cannot know that these pixels are not influenced by lower defoliation levels. Slightly lower NDVI values may be due to meteorological conditions, but also due to minor defoliation by small larvae populations. Since it is not possible to distinguish between the two causes in remote sensing data such an analysis would increase uncertainties. Instead we modeled GPP based on PAR for the five years for which we have data from the EC-tower and compared to measured GPP. A comparison between measured GPP and PAR-modeled GPP suggests that the birch forests were slightly defoliated by growing larvae populations the two years prior to the outbreak in 2012. In 2007 and 2009 measured and PAR-modeled GPP agreed well. In 2007, measured GPP was slightly lower than PAR-modeled GPP and in 2009 measured GPP was slightly larger than PAR-modeled GPP. We have also observed lower NDVI values in the years prior to an outbreak in time-series of NDVI over the study area. One important note here is that the annual GPP values in Table 1 (p. 15 in manuscript) were not displayed in the correct order for the years 2009, 2010, 2011 and 2014. The correct numbers in Table 1 are given below and agree with the lower annual GPP values in 2010 and 2011:

| Year | Years without insect outbreak | | | | | Outbreak |
|---|---|---|---|---|---|---|
| | 2007 | 2009 | 2010 | 2011 | 2014 | 2012 |
| **GPP** (g C m$^{-2}$ yr$^{-1}$) | 451 | 531 | 373 | 401 | 448 | 180 |

For the outbreak year 2012 the difference between EC derived GPP and PAR-modeled GPP was 286 g C m$^{-2}$ yr$^{-1}$, which is close to the decrease of 261 g C m$^{-2}$ yr$^{-1}$ estimated in the study. In addition, we ran the LUE model with meteorological data from the scientific research station in Abisko (ANS) for the year 2008 to fill the gap in the time-series with measured GPP and to study how well it agreed with the years 2007 and 2009. According to the LUE model the annual GPP at the EC tower was 440 g C m$^{-2}$ yr$^{-1}$ in 2008, which indicates that the GPP for undisturbed years of 441 g C m$^{-2}$ yr$^{-1}$ that we use is a reasonable value. We have added a discussion about these uncertainties and the challenge to find baseline conditions for GPP in areas with reoccurring insect outbreaks. In addition, a figure showing EC-derived and PAR-modeled GPP was added to the supplementary material. We have also added references that have found that the birch forests appear to reach pre-outbreak LAI and GPP 2-3 years after an outbreak (Hoogesteger & Karlsson 1992; Michal 2012).

As a response to the first part of the comment above we studied correlations between NDVI and meteorological data available from ANS, where we used the mean of the highest seasonal NDVI$_{DL}$ value derived from 200 MODIS pixels with birch forest. To minimize the influence of insect induced defoliation we excluded the outbreak years and years immediately prior to and after outbreaks. No linear correlations between PAR and GPP were found. There were, however, negative correlations between temperature and NDVI$_{DL}$, with the strongest correlation between NDVI$_{DL}$ and the mean temperature in May-June. The influence of temperature on NDVI$_{DL}$ was however, weak. Due to the low influence on NDVI and the large estimated uncertainty of the LUE model (30%) we did not adjust for these correlations in the analysis. We do, however, mention these results in the discussion but due to the limited amount of data we do not further elaborate on the results as that would be speculations.

There are studies related to insect outbreaks and climate but results are partly contradicting and only weak correlations between climate variables and outbreaks are found (se e.g. Young et al. 2014 and references within). Hence, we did not include this in the manuscript.

*2. In the Discussion, the authors must at least explicitly acknowledge four major methodological weaknesses; when possible, explain the likely impact of each weakness (i.e., under- or overestimating defoliation-caused GPP losses) and propose a way to address the weakness. (1) fAPAR was based on measurements for the upper canopy only, so it is unclear to which extent the fAPAR vs. NDVI_DL relationship applies to the entire forest. This is particularly critical due to the (possible) growth release of the understory highlighted by the authors. (2) The fAPAR vs. NDVI_DL relationship was derived for undefoliated years (2010-2011) only, yet was also used in the GPP_lue,defoliated model. Why not developing a fAPAR vs. NDVI_DL for defoliated conditions (no defoliation event at the spectral tower over the entire study)? (3) The defoliation detection algorithm missed 26% of defoliated areas and misclassified 39% of undefoliated areas. (4) How representative was the EC tower footprint of the entire study area, both during outbreak and no-outbreak years? This is critical, as EC tower data provided the basis for all GPP estimates through the values of nepsilon_max, epsilon_max,def, and the GPP reduction factor.*

Response: We agree that these weaknesses need to be discussed and have added them as limitations in the discussion: (1) This limitation is not easily handled since considerably more data are required to derive fAPAR and NDVI relationships depending on understory responses. We have, however, clarified the limitation in the discussion mentioning that a model including different relationships between fAPAR and NDVI depending on understory responses will be complex. (2) Unfortunately, the larvae were disrupting the PAR-sensors during the outbreak; hence, we have no reliable fAPAR data for defoliated conditions. We have added a section about this limitation in the discussion. (3) We have added a section about the accuracy of the defoliation detection method in the discussion: "The accuracy of the defoliation detection method also influences the results of the study. The method missed 26% of the defoliated MODIS pixels and misclassified 39% of the undisturbed pixels as defoliated in the evaluation data used by Olsson et al. (2016). This implies that the defoliated areas in 2004 and 2013 were slightly overestimated, while the defoliated area in 2012 was likely underestimated. However, the impact on the total numbers is likely small. It should also be considered that 20% of the forests in the study area were excluded since they are located in MODIS pixels with < 50% forests cover. The detection accuracy of the method may also influence the spatial distribution of the defoliation in the resulting maps, but even though there is an associated uncertainty at pixel level the broader outbreak patterns are likely reliable." (4) We agree that this could be an important limitations but according to Heliasz (2012) GPP is relatively stable over the study area. We have clarified this in the discussion: "There are also uncertainties in how well the EC tower footprint represents the study area. Heliasz (2012) utilized a permanent EC tower as reference and a mobile EC tower to study variability in carbon exchange in the birch forests around Abisko and concluded that there were only minor differences in GPP at seven sites during the peak growing season in 2008 and 2009. Hence, we consider the EC-tower footprint to be representative for the study area."

*3. I object to providing the 3-year \*total\* GPP reduction caused by defoliation, as this inadequately inflate numbers. Please provide the 3-year \*mean\* reduction instead throughout the text, making it clear the reduction is for outbreak years only (not the mean values over all years since 2000): Abstract; P16, L15 to P17, L1; P17, L7-8; P19, L5-7; P19, L15; P21, L18-20.*

Response: We agree that the total reduction in GPP may inflate the numbers. Hence, we have updated the manuscript accordingly except for in the discussion where we want to keep the comparison between the total decrease in GPP for the three outbreak years with the mean annual GPP for years without defoliation. We did, however, clarify that the total reduction was for the three years: "The total decrease in regional GPP, due to the three insect defoliation events studied here was estimated to be 44.6 ±13 Gg C, which is of the same magnitude as the average annual regional GPP of 41.1 ±12 Gg C yr$^{-1}$ for single years with no disturbances."

*4. P3, L15-26. I have various issues with the text from "Since near-linear" to "(Liljedahl et al. 2011)". First, it should be in the Methods; I suggest merging at the beginning of Section 2.3. Second, units are not provided for the variables and the equation is not numbered; please number \*all\* equations and provide units for \*all\* variables throughout the text (even when unitless; e.g., NDVI). Third, the sentence starting on L20 is cumbersome; if kept in Section 2.3, I suggest re-writing along the following lines: "The light use efficiency coefficient varies between vegetation types and the influence of meteorological conditions is accounted for through reduction factors for temperature and vapour pressure deficit [...]". Fourth, the sentence starting on L23 is an overstatement because: 1) temperature is not*

*always the main limiting factor in cold climates (Nemani et al. 2003; Beer et al. 2010); and 2) neither Bergh et al. (1998) nor Lagergren et al. (2005) really tested for the impact of factors other than temperature in their studies that covered only two sites in Sweden. The authors should acknowledge that they assumed accounting for temperature only was sufficient in their study region, an assumption supported (but not demonstrated) by Bergh et al. (1998) and Lagergren et al. (2005). Fifth, delete the sentence starting on L25: water stress is a major limiting factor in some boreal and other forests, not just for "ecosystems dominated by non-vascular plants".*

Response: We have moved most of the section to Methods as suggested and re-written or removed some parts of the text. What remains in the Introduction is: "Since near-linear relationships between satellite derived vegetation indices and the fraction absorbed PAR (fAPAR) have been established (e.g. Asrar et al. 1984; Sellers 1987; Goward & Huemmrich 1992; Myneni & Williams 1994; Olofsson and Eklundh 2007), it is possible to create a LUE model driven by remote sensing data. Such a LUE model could be applied for...".  Consequently, Section 2.3 is updated according to the suggestions. The equation and the sentence "ecosystems dominated by non-vascular plants" are removed. We have also added to the methods (Section 2.3) that: "We assumed that accounting for temperature only is sufficient in our study region, which is supported by Bergh et al. (1998) and Lagergren et al. (2005)."

*5. P6, L12. Why wasn't fAPAR_canopy also smoothed with TIMESAT before the regression?*

Response: $fAPAR_{canopy}$ used in the regression is the mean of daily $fAPAR_{canopy}$ over eight day periods, coinciding with the MODIS eight day periods. Hence, the $fAPAR_{canopy}$ values in the regression are already smooth and we did not want to risk introducing further artefacts by applying more smoothing. However, we have clarified in section 2.2.2 that we are working with eight day mean values: "Average $fAPAR_{canopy}$ over eight day periods, coinciding with the MODIS eight day periods, were computed and an ordinary least squares (OLS) regression was performed...."

*6. P7, Figure 3. The TIMESAT smoothing removed a second NDVI 'peak' each year. Please explain what was the origin of this (wintertime?) second annual peak and why removing it was OK.*

Response: The second peak occurs during the winter when there is no vegetation in the area. We have not studied the origin of the second peak which could be due to e.g. snow or darkness (no light in the winter season). Hence, we do not discuss the second peak, but we have added a clarification to the figure caption explaining that removing the second peak is OK: "There is small peak in raw NDVI (left) appearing each year. This peak appears during the winter when there is no vegetation in the study area and is hence, removed from the smoothed data (right)." (Left and right will be replaced by a and b according to comment #27).

*7. P12, L4-6. I do not understand why Method 1 should not also capture the effect of refoliation: the EC tower data should account for the post-refoliation increase in GPP, no? Unless no refoliation occurred within the EC tower footprint after the 2012 outbreak?*

Response: To our knowledge there was no refoliation around the EC tower in 2012, hence, the reduction factor represents an outbreak year without refoliation.

*8. P14, Figure 6. Many readers will likely expect defoliation to substantially decrease NDVI (due to a much lower leaf area index (LAI)) and leave LUE barely affected, so they will question the defoliation results (small NDVI decrease, large LUE decrease). It would thus be helpful to explain that small reductions in NDVI are associated with large reductions in LAI (e.g., Wulder et al. 1998), while LUE can*

*substantially decrease for lower LAI because more leaves operate in the light-saturated portion of the photosynthesis curve (e.g., Medlyn 1998). Also, please indicate the weeks during which defoliation occurred.*

Response: We agree with this reasoning and have added a section to the discussion: "It may seem unexpected that the difference in $NDVI_{DL}$ between undisturbed and defoliated years was comparably low in relation to the difference in light use efficiency. Mean $NDVI_{DL}$ for the peak of the growing season was 0.78 for the five years with EC-data. In 2012 the highest value for $NDVI_{DL}$ was 0.63. The difference in maximum light use efficiency was larger with an $\varepsilon_{max}$ of 1.85 ±0.36 g C $MJ^{-1}$ for years without disturbance, and an $\varepsilon_{max, def}$ of 0.98 ±0.25 g C $MJ^{-1}$ during defoliation. It is however, well known that NDVI saturates for high LAI and that small changes in NDVI can be associated with large changes in LAI (e.g. Myneni et al. 2002). The light use efficiency on the other hand can decrease substantially with lower LAI since more leaves will operate in the light-saturated portion of the photosynthesis (e.g. Medlyn 1998)." Defoliation usually occurs just after budburst (second part of June) and refoliation in late July or early August.

*9. P15, Figure 7 and P16, Figure 8. Give the equations for the regression lines on Figures 7 and 8. The line seems pretty close to 1:1 with zero intercept in Figure 7 (hence no bias for no-outbreak years), but the GPP_lue,defoliation model in Figure 8 seems to underestimate GPP over most of the range of actual values (i.e., up to ˍ2 gCˆm-2 dayˆ-1); this should be added to the list of weaknesses in the Discussion (see comment #2).*

Response: We have added the equations for the regressions line in the figure captions. For non-outbreak years the equation is: $GPP_{lue}$ = -0.12 × 1.01GPP. We also added the following to the text: "The intercept is -0.11 and the slope is 1.01 indicating that there is no bias in LUE modelled GPP for years without outbreaks." in Section 3.2, and "The figure, with an intercept of -0.54 and a slope of 1.25 indicates that the LUE model underestimates GPP for lower values." in Section 3.3.1. Furthermore, we will elaborate on the topic as a weakness in the discussion: Figure 8 suggests that the LUE model for defoliation underestimates GPP for values lower than about 1.5 g C $m^{-2}$ $day^{-1}$. This is a potential limitation of the LUE model developed for years with defoliation and would require EC-data from more years with defoliation to study. However, for this specific year the underestimated values from the LUE model are mainly due to a cold spring that resulted in a large reduction factor ($f_{8day}$). During the main growing season LUE modelled and EC-derived GPP agree well.

*10. P18, L7-10. Unless I am mistaken, the authors do not correctly interpret these results.If Method 2 captured the effect of refoliation whereas Method 1 did not, then Table 3 results should have a higher absolute value for refoliated than non-refoliated pixels \*regardless\* of the actual sign. This is what was obtained, suggesting that Method 2 did better capture refoliation's effect; however, the differences between refoliated and non-refoliated pixels are always within the error margin, so this better performance is marginal. (The sign of Table 3 values (negative for 2004 and 2012, positive for 2013) is not directly related to the effect of refoliation, but to the bias between the two methods.)*

Response: It is true that we made a mistake for 2013 when there is little difference between the two methods. Thanks for noting this. We have updated the manuscript accordingly: "For all years the mean differences in GPP loss (g C $m^{-2}$ $yr^{-1}$) between the methods were lower for pixels that recovered later in the growing season. These results suggest that Method 2 captured some of the refoliation, though the differences are small and within the error margin."

*11. P19, L26 to P20, L3. I do not think these explanations for Table 3 results (i.e, the difference in GPP loss between Method 1 and Method 2) are appropriate. The refoliation argument does not hold because, as noted by the authors, both 2004 and 2013 had high refoliation yet had opposite signs for Method 1 minus Method 2. Furthermore, the lower GPP losses in 2013 for Method 2 were basically the same for non-refoliated and refoliated pixels (Table 3). The argument of "uncertainties in nepsilon_max,def" does not seem appropriate either, because the same value was used for all years (so why a sign change in 2013 for Method 1 minus Method 2?). The argument of higher NDVI due to understory growth would work if this growth was stronger in 2013 compared to 2004 and 2012; are there reasons to believe this was the case? Here is another hypothesis that could account for the much lower mean GPP reduction in 2013 under Method 2 compared to all other values (and hence account for the sign change in 2013): could it be that growing conditions were better in 2013, thereby leading to higher f_8day and/or PAR_8day used in GPP_lue,defoliated compared to the mean f_8day/PAR_8day no-outbreak values used in GPP_lue?*

Response: We do agree that the discussion in this paragraph was a bit weak. We have removed the part about refoliation and uncertainties in $\varepsilon_{max,\ def}$. We studied meteorological data and compared the seasonal development in NDVI for the years 2004 and 2013. The seasonal trajectories of NDVI suggest that the growing season was shorter and that refoliation started earlier in 2013, which is one possible explanation for the smaller decrease in annual GPP for Method 2. We have added this to the discussion: "For the years 2004 and 2012, the two methods resulted in similar estimates of the GPP loss with slightly larger decrease in GPP for Method 2. In 2013, the difference between the methods was larger with the highest decrease in annual GPP for method 1. One possible explanation for the smaller decrease in annual GPP according to Method 2 for the year 2013 is that the growing season seems to have been shorter and that refoliation started earlier and was stronger in 2013 compared to 2004; this is indicated by the seasonal developments of NDVI for the pixels around the EC-tower."

**Minor comments:**
We agree with most of the minor comments. To keep this response short we only include the minor comments that we want to give any specific response to. For all minor comments that are not listed below the manuscript has been updated accordingly.

*12. Throughout the text: when possible replace the vague "carbon uptake" expression by the more accurate term applicable (GPP, NEE, etc.). Therefore, the title should be: "Mapping the reduction in gross primary productivity due to insect defoliation in subarctic birch forests". In the Discussion, the authors should stress that their results are not for the net carbon balance, but for GPP only; based on Heliasz et al. (2011), would it be possible to speculate whether, on a percentage basis, NEP losses should be higher or lower than GPP losses?*

Response: We have changed to either GPP or NEE and accordingly also changed the title as suggested and we have stressed that the results are for GPP only. During the outbreak in 2012 the decrease in $R_{eco}$ was much larger than decrease in GPP during the growing season around the EC tower.

*15. P2, L1. Reference(s) should support the "a warmer climate is likely to increase forest productivity" part of the sentence. Here are some suggestions: Pastor and Post (1988), Nemani et al. (2003), or Boisvenue and Running (2006).*

Response: We have added "(e.g. Nemani et al. 2003; Boisvenue & Running 2006)" as suggested.

*19. P3, L27-28. Bright et al. (2013) already used remote sensing (Landsat to identify trees killed by the mountain pine beetle) and a LUE model (MODIS GPP results, which*

*are based on LUE) to quantify the impact of an insect outbreak on carbon uptake. To my knowledge, the authors can still claim being the first ones to do it for defoliators.*

Response: We have updated the manuscript accordingly and added Bright et al. (2013) as a reference.

*23. P5, L18 to P6, L1. The purpose of "quality classes based on QA data" is not clear: how where these classes used in the analysis?*

Response: We agree that mentioning the quality classes was confusing, hence, we have changed the text: "Double logistic functions were used to smooth the raw NDVI data and QA data from both MOD09Q1 and the more comprehensive QA-flags in MOD09A1 were utilized to estimate the quality of the NDVI observations."

*25. P6, L8. Is the "pure canopy absorbed PAR" different from fAPAR_canopy? If yes, quickly explain what is the difference and why it matters. If not, replace the expression by "fAPAR_canopy".*

Response: Pure canopy absorbed PAR is the same as $fAPAR_{canopy}$. We have updated the manuscript accordingly.

*27. P7. I suggest combining Figure 2 with Figure 1. For all multi-panel Figures, please add letters to each panel and refer to the appropriate panel in the text (e.g., Fig. 5a).*

Response: We do agree that it could be a good idea to merge the figures so that the orthophoto over the area around the EC-tower is shown together with the location of the tower. However, we decided to keep them as separate figures with the main motivation that we want to keep the size of Figure 2 large to make the photo easy to interpret. Letter have been added to multi-panel figures.

*32. P10, L3. T_mean8 was never negative for such a northern site at the end of September? If negative values occurred, explain how they were dealt with (Equation (8) suggests a negative value for f_8day, which would make no sense).*

Response: There were no negative values for $T_{mean8}$ in the period studied (coldest value was 3°C). There were eight day periods with temperatures below zero, but no eight day period with a mean value < 0°C.

*36. P11, L15-16. The text should be expanded, because at this point the reader is still unaware that an outbreak occurred in 2012 within the EC tower footprint: please state this explicitly here and add what was the percentage of defoliation within the tower footprint.*

Response: We have clarified according to the comment: "(1) a method based on a reduction factor derived from the EC data from 2012 when the birch forest in the footprint of the tower was severely defoliated, and no refoliation occurred. This reduction factor was applied to all pixels in the study area"

*48. P21, L1-2. For (partially) contrasting modeling results about the effect of insect outbreaks on the carbon balance, see Seidl et al. (2008), Albani et al. (2010), and Landry et al. (2016) (the last one is not the same reference as already cited in the manuscript).*

Response: Thanks for these interesting references. We did not include them though as they mainly discuss longer term impacts.

*50. P21, L13-14. Delete the sentence: there is no firm basis for such an extrapolation, and the number would then likely end up being cited by future studies...*

Response: Since these insect infestations occur frequently across very large areas in the region we think that it is relevant to include some information about the potential effect on the carbon uptake. However, we agree that it is an uncertain estimation and reformulated the text accordingly: "Assuming that the conditions were similar over northern Fennoscandia, the insect defoliation over these vast areas would result in a potential total regional GPP loss for the time period of the magnitude 2–3 Tg C."

***References:***

Heliasz (2012). PhD thesis, Dep. of Geography and Ecosystem Science, Lund Univ., Lund, Sweden.

Hoogesteger and Karlsson (1992). Functional Ecology, 6, 317-323, 10.2307/2389523.

Young et al. (2014). Arctic, Antarctic, and Alpine Research, 46, 659-668, 10.1657/1938-4246-46.3.659.

---

## Author Comment (AC2) · 12 Jan 2017

Dear Referee two. We thank you for the constructive and detailed comments and suggestions. We agree with most of the comments and have made some further analysis to improve the study. We have made most of the required updates to the manuscript.

***Specific comments:***

*In several places the approach seems to take too simplistic a view, without properly discussing the assumptions or their impact. For example, a key result presented is the difference between GPP derived from EC measurements in an outbreak year (2012) compared to five other years without insect utbreaks. Unfortunately, it seems EC data were only available for one outbreak year, but there is no analysis of differences between years due to factors other than insect damage. Inter-annual variability in meteorological conditions (rainfall/soil moisture, solar radiation and temperature in particular) can result in different annual total GPP. The values given in Table 1 should therefore be analysed with respect to meteorological conditions. This should also allow the 2012 value to be given some context – if the insect outbreak had not occurred, would the 2012 total be lower/higher/similar to the average based on meteorological conditions alone? Furthermore, is it possible that the outbreaks in 2012 and 2013 contributed to the lower GPP obtained in 2014, or is this attributable to meteorological conditions?*

Response: We agree that these important considerations need to be discussed, and we have consequently studied relationships between meteorological conditions and GPP as well as $NDVI_{DL}$. One important note here is that the annual GPP values given in Table 1 (p. 15 in manuscript) were not displayed in the correct order for the years 2009, 2010, 2011 and 2014. The correct figures in Table 1 are:

| | Years without insect outbreak | | | | | Outbreak |
|---|---|---|---|---|---|---|
| Year | 2007 | 2009 | 2010 | 2011 | 2014 | 2012 |
| **GPP** (g C m$^{-2}$ yr$^{-1}$) | 451 | 531 | 373 | 401 | 448 | 180 |

where annual GPP was largest in 2007 and 2009, and the lower annual GPP in 2010 and 2011 indicates that there were minor defoliation already in these years.

Due to the limited number of years with EC derived GPP we did not consider it reliable to study correlations between annual GPP measured at the EC tower and meteorological variables. Instead we modeled GPP for the birch forest around the tower with PAR, and compared EC derived and PAR-modeled GPP. The comparison suggests that in the two years (2010 and 2011) prior to the outbreak, measured GPP was lower than PAR-modeled GPP, indicating that there were signs of defoliation by growing larval densities. Also in 2014 when the birch forest most likely was recovering from the outbreak, measured GPP was lower than PAR-modeled GPP. For the earlier years (2007 and 2009) when the birch forest was likely closer to undisturbed conditions, EC derived GPP and GPP modeled with PAR data agreed well; measured GPP was slightly lower compared to PAR-modeled in 2007 and slightly larger in 2009. For the outbreak year 2012 the difference between EC derived GPP and PAR-modeled GPP was 286 g C m$^{-2}$ yr$^{-1}$, which is similar to the decrease of 261 g C m$^{-2}$ yr$^{-1}$ we found in our study. In addition, we ran the LUE model with meteorological data from the scientific research station in Abisko (ANS) for the year 2008 to fill the gap in the time-series with measured GPP and to study how well it agreed with the years 2007 and 2009. According to the LUE model the annual GPP at the EC tower was 440 g C m$^{-2}$ yr$^{-1}$ in 2008. This indicates that the GPP for undisturbed years of 441 g C m$^{-2}$ yr$^{-1}$ that we used is reasonable.

However, with data from the EC tower available for more years it would be a potentially important improvement to include meteorological data when estimating the decrease in annual GPP. We have added a section about these results in the discussion and the figure showing EC derived and PAR-modeled GPP was added to the supplementary material.

We also studied correlations between NDVI and meteorological data available from ANS, where we used mean of the highest seasonal $NDVI_{DL}$ value derived from 200 MODIS pixels with birch forest. To minimize the influence of insect induced defoliation we excluded the outbreak years and years immediately prior to and after outbreaks. No linear relationships between PAR and GPP were found. There were, however, negative correlations between temperature and $NDVI_{DL}$, with the strongest correlation between $NDVI_{DL}$ and the mean temperature in May-June. The influence of the temperature on NDVI was however, weak and due the large estimated uncertainties of the LUE model (30%) we did not include these correlations in the analysis. We do, however, mention these results in the discussion but due to the limited amount of data we do not further elaborate on the results as that would be speculation.

*The comparison with the 2004 results of Heliasz et al. (2011) on page 20 is useful and indicates that closer analysis of the temporal evolution of the EC data may be beneficial. Currently data are separated into years with and without insect outbreaks. During those years with outbreaks, does the reduction in GPP over the course of the growing season agree with the timing of insect population growth/insect damage? Is this also supported by the NDVI data? In the years classified as being without insect outbreaks, are there any effects of (albeit smaller) insect populations on the GPP or NDVI values? Perhaps such analyses could offer further insight into the refoliation effect; it is currently hard to draw meaningful conclusions on this subject from the information given in the article. Some evidence to support the assumption that NDVIDL captures refoliation would also be useful; Fig 3 is not very convincing in this respect.*

Response: For the year 2004, when we had EC data for the later part of the insect defoliation and the following refoliation at the EC tower, we can see that GPP is low during the defoliation events and increasing later in the growing season with the new leaves appearing. The raw NDVI values have a similar pattern with lower values (around 0.6) until early August when refoliation results in a late season peak in NDVI. This is illustrated in Figure 3 where raw NDVI has a similar seasonal development in 2013, when there was substantial refoliation around the EC tower: raw NDVI stayed around 0.6 during June, but increased to pre-outbreak levels in early July when refoliation occurred. In 2012 when there was no refoliation around the EC tower, raw NDVI stayed around 0.6 during the entire growing season. These different seasonal trajectories are utilized in the defoliation detection method (Olsson et al. 2016). $NDVI_{DL}$ does, however, not capture the typical trajectory for refoliation years with sharply increasing NDVI values when the growing season starts, that level off and start increasing again later in the season. The higher NDVI values in the later part of the growing season result in $NDVI_{DL}$ values that are higher than in 2012 but still lower than for years without defoliation (even though the actual timing of the defoliation is not captured in the LUE modeled GPP during years with refoliation). A new version of TIMESAT, currently being developed and tested, will capture also more detailed seasonal trajectories with smooth fitting of curves. These new curve fitting methods have a potential to improve the performance of the LUE model. We have added a section about our response to this comment to the discussion.

As mentioned in our previous comment we found influence on GPP at the EC tower the two years prior to the outbreak that are likely due to increasing insect population before reaching outbreak levels. This pattern is suggested in time-series of NDVI where there seems to be weak signs of defoliation 1-2 years prior to the outbreak.

*P15, L6-10: Section 2.2.3 states that EC data were available from 1 May to 30 Sep, covering most of the growing season. Do the EC observations and values given in Table 1 agree with this timeframe, or is it possible GPP in Table 1 is underestimated if the growing season extended beyond these dates? This study focuses on GPP, but could the authors comment on other potential impacts of the insect outbreak on the carbon balance? For example, how might respiration rates be affected, and might this impact the partitioning into Reco and GPP?*

Response: Budburst usually occurs early in June or very late in May so including data from first of May means that we capture the start of the growing season. For the years included in this study GPP was approaching zero by the last week of September implying that there were no underestimates of annual GPP.

Respiration is affected by insect outbreak in two ways: (1) Autotrophic respiration is reduced as defoliated trees cannot photosynthesize and (2) heterotrophic respiration increase when dead larvae decompose. The amount of carbon respired by larvae should be the same as the amount of carbon in eaten leaves so we should only observe a shift of respiration in time. In addition, larvae move nutrients from trees to fungi and bacteria living in soil which further increase respiration. The increase in heterotrophic respiration did not offset decrease in autotrophic respiration and $R_{eco}$ for outbreak year was decreased in comparison to non-disturbed years.

*On a similar note, many of the decisions taken in the presentation of results and development of the model rely on data collected during non-outbreak years. Is the gap-filling approach also suitable in defoliated years?*

Response: The gap-filling approach is suitable also for defoliated years since the gap-filling function is created based on data from short time windows, usually seven days, even though the window can be longer if there are insufficient data for correct fitting. This short time window adjusts the fitting parameters for changing ecosystem conditions. We have clarified this in the manuscript.

*P8, L10-11: More detail is needed about the quality control. Under which 'bad' weather/measuring conditions were data removed? How much data remained after quality control and what proportion was gap-filled?*

Response: Data were removed mainly due to precipitation since an open path gas analyzer was used, or if the atmosphere was not fulfilling the turbulent conditions required for eddy covariance measurements. Available data after cleaning: 2007 61%; 2009 71%; 2010 66%; 2011 65%; 2012 58%; 2014 65%. We have clarified this in the manuscript.

*P10, L11-2: Here, it is not clear which years have been used or why. The years for which the EC data are available should be stated in Section 2.2.3. Why was 2012 the only year used to calculate "max with insect defoliation? Why were data from 2008 and 2013 not included/not available?*

Response: We have clarified that EC data from undisturbed years are from 2007, 2009, 2011, 2012 and 2014. In 2008 data were missing due to instrument failure. In 2013 the measurements were disturbed by larvae which unfortunately leave us with EC data from one year only.

*P12, L13: It is not clear how these statistics were calculated and they don't seem to follow from Fig 4. Please provide more details/clarification*

Response: We do agree that these statistics were not well described. Previously the statistics were based on the entire study area. We have changed the statistics to include only the pixels around the EC tower to correspond to the figure and clarified this. "The influence of observations with $NDVI_{DL}$ values < 0.4 and with $f_{8day}$ > 0 was small. For the years with data available from the EC tower 8% of the eight day periods had $NDVI_{DL}$ < 0.4 and $f_{8day}$ > 0 in the MODIS pixels surrounding the tower. For these time periods average $f_{8day}$ was 0.068."

*P14, Fig 6: The two green lines for NDVIDL show higher NDVI values early in the growing season for the defoliated year. Possible reasons for this should be investigated and the impact on the results commented on.*

Response: Thanks for pointing this out. This is due to a weak fitting of the double logistic functions in TIMESAT. In a currently developed version of TIMESAT the fitting of the functions will be more robust. We have added a clarification in Section 3.3.1:"In Figure 6, $NDVI_{DL}$ has higher values in the year with defoliation compared to undisturbed years in May (period 16-18). These high $NDVI_{DL}$ values are due to poor fitting of the double logistic function during winter and early spring in 2012 (see Figure 3, where $NDVI_{DL}$ increases earlier in 2012 compared to the other years). The impact on the result is however small since these eight periods are in the early part of the growing season and the reduction factor ($f_{8day}$) is zero."

**Minor comments:**
We agree with most of the minor comments. To keep this response short we only include the minor comments that we want to give any specific response to. For all minor comments that are not listed below the manuscript has been updated accordingly.

*P2, L26: Would be useful to give the land cover type for the southern France site*
Response: We have added that the defoliation occurred in holm oak (Quercus ilex L.).

*P3, L17: Change to read 'of the form'*
*P3, L18: Number this equation and update the others accordingly*
*P3, L20-1: Change to read 'and with variability in meteorology'*
Response: The section is updated and these comments are no longer relevant.

*P4, L24-6: It is not clear that these recent outbreaks are for the study site – please state the area they apply to. Also give some information about the EC tower in that study (i.e. location, and mention the flux measurements were also for the birch forest)*
Response: We have clarified that the outbreaks mentioned were in the study area, and that the EC tower mentioned was located in birch forest: "The latest outbreaks in the study are occurred in 2004, with a documented reduction in carbon sink strength of 89% at an EC tower located in birch forest…"

*P11, L22-23 Suggest choosing alternative notation for GPPreduction, as it is a ratio of GPPs, rather than GPP itself*

Response: We have changed the notation of the reduction factor to *GPPredfact*

*P14 L5-6 and Fig 7: The text mentions 'low GPP observations' but in Fig 7 it looks as though the modelled GPP values are lower than the EC observed values, with several zero values. Please clarify*

Response:  The low GPP observations with several zero values for LUE modelled GPP are from May, before budburst for the birch forest, when there is photosynthetic activity in understory captured in the EC data. These low GPP values have little influence on annual GPP. We have added a note about this in the discussion but do not further elaborate in potential causes since we have not studied this.

---

## Author Response (AR1)

Dear Editor,

We thank you and the referees for all comments that have helped us to improve the manuscript. We have updated the manuscript according to your last comment:

5 We have changed unbiased to (P. 14, L. 3-4): "The intercept is -0.11 and the slope is 1.01 indicating that the LUE model performs well for years without outbreaks." We have also changed to two significant digits in all results.

In this document we include detailed responses to Referee 1 (P. 2-11) and Referee 2 (P. 12-16) followed by a marked-up version of the manuscript (P. 17-51).

10 Sincerely,

Per-Ola Olsson

Dear Referee one. We thank you for the constructive and detailed comments and suggestions. We agree with most of the comments and have made some further analysis to improve the study. We have made most of the required updates to the manuscript.

**Major comments:**

*1. Using the mean GPP over no-outbreak years as the basis to estimate GPP reduction caused by outbreaks seems inadequate for two different reasons. First, differences in weather may affect both GPP and outbreak occurrence/severity. If the two variables are indeed correlated across years, the approach likely causes a bias. This should be checked; no need for fancy statistical tests, just compare, based on the GPP_lue model, the*
10 *mean GPP and its standard deviation for outbreak vs. no-outbreak years over pixels that have never been defoliated. Second, GPP in pixels previously defoliated is unlikely equal to what it would have been if no outbreak had occurred. For example, canopy trees might have not fully recovered yet (hence underestimating the no-outbreak GPP) or total tree+understory productivity might increase for a few years due to the defoliation-caused growth release of the understory (hence overestimating the no-outbreak GPP). The authors should rather have*
15 *estimated the no-outbreak GPP in pixels that have been defoliated up to X years before (with X to be defined; maybe 3-5 years?) based on the NDVI_DL values of neighbouring pixels that have never been defoliated. (More precisely: for each defoliated pixel, define a window large enough to include never-defoliated pixels, but small enough to have similar conditions. Then, compute the mean NDVI_DL difference between the defoliated pixel and the never defoliated pixels over all years prior to the (first) defoliation event in the defoliated pixel; let's say*
20 *NDVI_DL was on average 10% higher in the defoliated pixel. Finally, for the X years after the defoliation event (excluding the defoliation year itself), estimate the annual no-outbreak GPP in the defoliated pixel with the GPP_lue model, but with a value of NDVI_DL 10% higher than the mean annual NDVI_DL value over the neighbouring never-defoliated pixels [instead of using the annual MODIS NDVI_DL value in the defoliated pixel].) Ideally, the authors should re-do their analyses using this new approach. At a minimum, the authors must use*
25 *this new approach for >100 randomly-selected defoliated pixels and see to which extent it affects their results.*

Response: We agree that these are import points. We first respond to the issue with recovery after an outbreak and thank you for the good suggestion about how to handle this limitation. We did explore the possibility to perform the suggested analysis but decided not to go ahead with the analysis. The main motivation is the difficulty to find pixels without any defoliation.
30 Even though there are pixels that are not detected as defoliated during outbreak years we cannot know that these pixels are not influenced by lower defoliation levels. Slightly lower NDVI values may be due to meteorological conditions, but also due to minor defoliation by small larvae populations. Since it is not possible to distinguish between the two causes in remote sensing data such an analysis would increase uncertainties. Instead we modeled GPP based on PAR for the five years for which we have data from the EC-tower and compared to measured GPP. A comparison between measured GPP and PAR-
35 modeled GPP suggests that the birch forests were slightly defoliated by growing larvae populations the two years prior to the outbreak in 2012. In 2007 and 2009 measured and PAR-modeled GPP agreed well. In 2007, measured GPP was slightly lower than PAR-modeled GPP and in 2009 measured GPP was slightly larger than PAR-modeled GPP. We have also observed lower NDVI values in the years prior to an outbreak in time-series of NDVI over the study area. One important note here is that the annual GPP values in Table 1 (p. 15 in manuscript) were not displayed in the correct order for the years
40 2009, 2010, 2011 and 2014. The correct numbers in Table 1 are given below and agree with the lower annual GPP values in 2010 and 2011 (Note that we also changed to two significant digits):

| | Years without insect outbreak | | | | | Outbreak |
|---|---|---|---|---|---|---|
| Year | 2007 | 2009 | 2010 | 2011 | 2014 | 2012 |
| GPP (g C m$^{-2}$ yr$^{-1}$) | 450 | 530 | 370 | 400 | 450 | 180 |

For the outbreak year 2012 the difference between EC derived GPP and PAR-modeled GPP was 290 g C m$^{-2}$ yr$^{-1}$, which is close to the decrease of 260 g C m$^{-2}$ yr$^{-1}$ estimated in the study. In addition, we ran the LUE model with meteorological data from the scientific research station in Abisko (ANS) for the year 2008 to fill the gap in the time-series with measured GPP and to study how well it agreed with the years 2007 and 2009. According to the LUE model the annual GPP at the EC tower was 440 g C m$^{-2}$ yr$^{-1}$ in 2008, which indicates that the GPP for undisturbed years of 440 g C m$^{-2}$ yr$^{-1}$ that we use is a reasonable value. We have added a discussion about these uncertainties and the challenge to find baseline conditions for GPP in areas with reoccurring insect outbreaks (P. 20, L. 7-19). In addition, a figure showing EC-derived and PAR-modeled GPP was added to the supplementary material. We have also added references that have found that the birch forests appear to reach pre-outbreak LAI and GPP 2-3 years after an outbreak (P. 20, L. 3-5).

As a response to the first part of the comment above we studied correlations between NDVI and meteorological data available from ANS, where we used the mean of the highest seasonal NDVI$_{DL}$ value derived from 200 MODIS pixels with birch forest. To minimize the influence of insect induced defoliation we excluded the outbreak years and years immediately prior to and after outbreaks. No linear correlations between PAR and GPP were found. There were, however, negative correlations between temperature and NDVI$_{DL}$, with the strongest correlation between NDVI$_{DL}$ and the mean temperature in May-June. The influence of temperature on NDVI$_{DL}$ was however, weak. Due to the low influence on NDVI and the large estimated uncertainty of the LUE model (30%) we did not adjust for these correlations in the analysis. We do, however, mention these results in the discussion (P. 20, L. 21-32) but due to the limited amount of data we do not further elaborate on the results as that would be speculations.

There are studies related to insect outbreaks and climate but results are partly contradicting and only weak correlations between climate variables and outbreaks are found (se e.g. Young et al. 2014 and references within). Hence, we did not include this in the manuscript.

*2. In the Discussion, the authors must at least explicitly acknowledge four major methodological weaknesses; when possible, explain the likely impact of each weakness (i.e., under- or overestimating defoliation-caused GPP losses) and propose a way to address the weakness. (1) fAPAR was based on measurements for the upper canopy only, so it is unclear to which extent the fAPAR vs. NDVI_DL relationship applies to the entire forest. This is particularly critical due to the (possible) growth release of the understory highlighted by the authors. (2) The fAPAR vs. NDVI_DL relationship was derived for undefoliated years (2010-2011) only, yet was also used in the GPP_lue,defoliated model. Why not developing a fAPAR vs. NDVI_DL for defoliated conditions (no defoliation event at the spectral tower over the entire study)? (3) The defoliation detection algorithm missed 26% of defoliated areas and misclassified 39% of undefoliated areas. (4) How representative was the EC tower footprint of the entire*

*study area, both during outbreak and no-outbreak years? This is critical, as EC tower data provided the basis for all GPP estimates through the values of nepsilon_max, epsilon_max,def, and the GPP reduction factor.*

Response: We agree that these weaknesses need to be discussed and have added them as limitations in the discussion: (1)

5   This limitation is not easily handled since considerably more data are required to derive fAPAR and NDVI relationships depending on understory responses. We have, however, clarified the limitation in the discussion mentioning that a model including different relationships between fAPAR and NDVI depending on understory responses will be complex (P. 21. L. 1-8). (2) Unfortunately, the larvae were disrupting the PAR-sensors during the outbreak; hence, we have no reliable fAPAR data for defoliated conditions. We have added a section about this limitation in the discussion (P. 20, L. 31-34). (3) We have

10  added a section about the accuracy of the defoliation detection method in the discussion (P. 21, L. 21-26) (4) We agree that this could be an important limitation but according to Heliasz (2012) GPP is relatively stable over the study area. We have clarified this in the discussion (P. 21, L. 17-20): "There are also uncertainties in how well the EC tower footprint represents the study area. Heliasz (2012) utilized a permanent EC tower as reference and a mobile EC tower to study variability in carbon exchange in the birch forests around Abisko and concluded that there were only minor differences in GPP at seven

15  sites during the peak growing season in 2008 and 2009. Hence, we consider the EC-tower footprint to be representative for the study area."

*3. I object to providing the 3-year \*total\* GPP reduction caused by defoliation, as this inadequately inflate numbers. Please provide the 3-year \*mean\* reduction instead throughout the text, making it clear the reduction is for outbreak years only (not the*

20  *mean values over all years since 2000): Abstract; P16, L15 to P17, L1; P17, L7-8; P19, L5-7; P19, L15; P21, L18-20.*

Response: We agree that the total reduction in GPP may inflate the numbers. Hence, we have updated the manuscript accordingly except for in the discussion where we want to keep the comparison between the total decrease in GPP for the

25  three outbreak years with the mean annual GPP for years without defoliation. We did, however, clarify that the total reduction was for the three years (P. 19, L. 8-10): "The total decrease in regional GPP, due to the three insect defoliation events studied here was estimated to be $45\pm14$ Gg C, which is of the same magnitude as the average annual regional GPP of $41\pm12$ Gg C $yr^{-1}$ for single years with no disturbances.

*4. P3, L15-26. I have various issues with the text from "Since near-linear" to "(Liljedahl*

30  *et al. 2011)". First, it should be in the Methods; I suggest merging at the beginning of Section 2.3. Second, units are not provided for the variables and the equation is not numbered; please number \*all\* equations and provide units for \*all\* variables throughout the text (even when unitless; e.g., NDVI). Third, the sentence starting on L20 is cumbersome; if kept in Section 2.3, I suggest re-writing along the following lines: "The light use efficiency coefficient varies between vegetation types and the influence of meteorological conditions is accounted for*

35  *through reduction factors for temperature and vapour pressure deficit [...]". Fourth, the sentence starting on L23 is an overstatement because: 1) temperature is not always the main limiting factor in cold climates (Nemani et al. 2003; Beer et al. 2010); and 2) neither Bergh et al. (1998) nor Lagergren et al. (2005) really tested for the impact of factors other than temperature in their studies that covered only two sites in Sweden. The authors should acknowledge that they assumed accounting for temperature only was sufficient in their study region, an*

40  *assumption supported (but not demonstrated) by Bergh et al. (1998) and Lagergren et al. (2005). Fifth, delete the*

*sentence starting on L25: water stress is a major limiting factor in some boreal and other forests, not just for "ecosystems dominated by non-vascular plants".*

Response: We have moved most of the section to Methods as suggested and re-written or removed some parts of the text. What remains in the Introduction is (P. 3, L. 21-23): "Since near-linear relationships between satellite derived vegetation indices and the fraction absorbed PAR (fAPAR) have been established (e.g. Asrar et al. 1984; Sellers 1987; Goward & Huemmrich 1992; Myneni & Williams 1994; Olofsson and Eklundh 2007), it is possible to create a LUE model driven by remote sensing data. Such a LUE model could be applied for...". Consequently, Section 2.3 is updated according to the suggestions. The equation and the sentence "ecosystems dominated by non-vascular plants" are removed. We have also added to the methods (Section 2.3) that (P. 9, L. 12-13): "We assumed that accounting for temperature only is sufficient in our study region, which is supported by Bergh et al. (1998) and Lagergren et al. (2005)."

*5. P6, L12. Why wasn't fAPAR_canopy also smoothed with TIMESAT before the regression?*

Response: $fAPAR_{canopy}$ used in the regression is the mean of daily $fAPAR_{canopy}$ over eight day periods, coinciding with the MODIS eight day periods. Hence, the $fAPAR_{canopy}$ values in the regression are already smooth and we did not want to risk introducing further artefacts by applying more smoothing. However, we have clarified in section 2.2.2 that we are working with eight day mean values (P. 6, L. 11-12): "Average *$fAPAR_{canopy}$* over eight day periods, coinciding with the MODIS eight day periods, were computed and an ordinary least squares (OLS) regression was performed…."

*6. P7, Figure 3. The TIMESAT smoothing removed a second NDVI 'peak' each year. Please explain what was the origin of this (wintertime?) second annual peak and why removing it was OK.*

Response: The second peak occurs during the winter when there is no vegetation in the area. We have not studied the origin of the second peak which could be due to e.g. snow or darkness (no light in the winter season). Hence, we do not discuss the second peak, but we have added a clarification to the figure caption explaining that removing the second peak is OK (Fig. 3, P. 7): "There is a small peak in raw NDVI (a) appearing each year. This peak appears during the winter when there is no vegetation in the study area and is hence, removed from the smoothed data (b).

*7. P12, L4-6. I do not understand why Method 1 should not also capture the effect of refoliation: the EC tower data should account for the post-refoliation increase in GPP, no? Unless no refoliation occurred within the EC tower footprint after the 2012 outbreak?*

Response: To our knowledge there was no refoliation around the EC tower in 2012, hence, the reduction factor represents an outbreak year without refoliation. We have clarified this (P. 11, L. 23-24): "derived from the EC data from 2012 when the birch forest in the footprint of the tower was severely defoliated, and no refoliation occurred."

*8. P14, Figure 6. Many readers will likely expect defoliation to substantially decrease NDVI (due to a much lower leaf area index (LAI)) and leave LUE barely affected, so they will question the defoliation results (small NDVI decrease, large LUE decrease). It would thus be helpful to explain that small reductions in NDVI are associated with large reductions in LAI (e.g., Wulder et al. 1998), while LUE can substantially decrease for lower LAI because more leaves operate in the light-saturated portion of the photosynthesis curve (e.g., Medlyn 1998). Also, please indicate the weeks during which defoliation occurred.*

Response: We agree with this reasoning and have added a section to the discussion (P. 21, L. 13-16): "It may also seem surprising that the difference in $NDVI_{DL}$ was comparably low in relation to the difference in light use efficiency. It is, however, known that NDVI saturates for high LAI and that small changes in NDVI can be associated with large changes in LAI (e.g. Myneni et al. 2002). The light use efficiency on the other hand can decrease substantially with lower LAI since more leaves will operate in the light-saturated portion of the photosynthesis (e.g. Medlyn 1998)."

*9. P15, Figure 7 and P16, Figure 8. Give the equations for the regression lines on Figures 7 and 8. The line seems pretty close to 1:1 with zero intercept in Figure 7 (hence no bias for no-outbreak years), but the GPP_lue,defoliation model in Figure 8 seems to underestimate GPP over most of the range of actual values (i.e., up to _2 gC mˆ-2 dayˆ-1); this should be added to the list of weaknesses in the Discussion (see comment #2).*

Response: We have added the equations for the regressions line in the figure captions (Fig. 7, P. 15; Fig. 16, P. 8). We also added the following to the text (P. 14, L. 3-4): "The intercept is -0.11 and the slope is 1.01 indicate that the LUE model performs well for years without outbreaks." and (P. 16, L. 2-3) "The figure, with an intercept of -0.54 and a slope of 1.25 indicates that the LUE model underestimates GPP for lower values. Furthermore, we elaborate on the topic as a weakness in the discussion (P. 21, L. 8-12): "Another potential limitation is that the LUE model developed for years with defoliation seems to underestimate GPP for values lower than about 1.5 g C m$^{-2}$ day$^{-1}$ (Figure 8). However, for the outbreak year with available EC data (2012) the underestimated values from the LUE model are mainly due to a cold spring that resulted in a large reduction factor ($f_{8day}$). During the main growing season LUE modelled and EC derived GPP agrees well, which increases confidence in the modelling."

*10. P18, L7-10. Unless I am mistaken, the authors do not correctly interpret these results.If Method 2 captured the effect of refoliation whereas Method 1 did not, then Table 3 results should have a higher absolute value for refoliated than non-refoliated pixels \*regardless\* of the actual sign. This is what was obtained, suggesting that Method 2 did better capture refoliation's effect; however, the differences between refoliated and non-refoliated pixels are always within the error margin, so this better performance is marginal. (The sign of Table 3 values (negative for 2004 and 2012, positive for 2013) is not directly related to the effect of refoliation, but to the bias between the two methods.)*

Response: It is true that we made a mistake for 2013 when there is little difference between the two methods. Thanks for noting this. We have updated the manuscript accordingly (P. 18, L. 7-11): "We compared the differences in GPP decrease between Method 1 (GPP reduction factor) and Method 2 (two LUE models) to study if Method 2 performed better for MODIS pixels where the birch trees recovered later in the growing season. For all years the mean differences in GPP loss (g C m$^{-2}$ yr$^{-1}$) between the methods were lower for pixels that recovered later in the growing season. These results suggest that Method 2 captured some of the refoliation, though the differences are small and within the error margin.

*11. P19, L26 to P20, L3. I do not think these explanations for Table 3 results (i,e, the difference in GPP loss between Method 1 and Method 2) are appropriate. The refoliation argument does not hold because, as noted by the authors, both 2004 and 2013 had high refoliation yet had opposite signs for Method 1 minus Method 2. Furthermore, the lower GPP losses in 2013 for Method 2 were basically the same for non-refoliated and refoliated pixels (Table 3). The argument of "uncertainties in nepsilon_max,def" does not seem appropriate*

*either, because the same value was used for all years (so why a sign change in 2013 for Method 1 minus Method 2?). The argument of higher NDVI due to understory growth would work if this growth was stronger in 2013 compared to 2004 and 2012; are there reasons to believe this was the case? Here is another hypothesis that could account for the much lower mean GPP reduction in 2013 under Method 2 compared to all other values*

5 *(and hence account for the sign change in 2013): could it be that growing conditions were better in 2013, thereby leading to higher f_8day and/or PAR_8day used in GPP_lue,defoliated compared to the mean f_8day/PAR_8day no-outbreak values used in GPP_lue?*

Response: We do agree that the discussion in this paragraph was a bit weak. We have removed the part about refoliation and

10 uncertainties in $\varepsilon_{max, def}$. We studied meteorological data and compared the seasonal development in NDVI for the years 2004 and 2013. The seasonal trajectories of NDVI suggest that the growing season was shorter and that refoliation started earlier in 2013, which is one possible explanation for the smaller decrease in annual GPP for Method 2. We have added this to the discussion (P. 22, L. 26-30): "For the years 2004 and 2012, the two methods resulted in similar estimates of the GPP loss with slightly larger decrease in GPP for Method 2. In 2013, the difference between the methods was larger with the highest

15 decrease in annual GPP for method 1. One possible explanation for the smaller decrease in annual GPP according to Method 2 for the year 2013 is that the growing season seems to have been shorter and that refoliation started earlier and was stronger in 2013 compared to 2004; this is indicated by the seasonal developments of NDVI."

**Minor comments:**
*12. Throughout the text: when possible replace the vague "carbon uptake" expression by the more accurate term*
20 *applicable (GPP, NEE, etc.). Therefore, the title should be: "Mapping the reduction in gross primary productivity due to insect defoliation in subarctic birch forests". In the Discussion, the authors should stress that their results are not for the net carbon balance, but for GPP only; based on Heliasz et al. (2011), would it be possible to speculate whether, on a percentage basis, NEP losses should be higher or lower than GPP losses?*

25 Response: We have changed to either GPP or NEE and accordingly also changed the title as suggested and we have stressed that the results are for GPP only and we also mention in the discussion that (P. 19, L. 12-13): "during the outbreak in 2012 the decrease in $R_{eco}$ was larger than the decrease in GPP during the growing season around the EC tower."

*13. Throughout the text: please use the "regional GPP" expression when applicable, to make it clearer the values are for the total GPP over the study region.*
30
Response: We have updated the manuscript accordingly.

*14. P1, L18-20. I have various issues with this sentence; I suggest re-writing along the following lines: "In the study area of 100 km^2, the results suggested a mean regional*
35 *GPP decrease of XX +/- YY Gg C yr^-1 for the three outbreak years (2004, 2012, and 2013), compared to a mean regional GPP of 41.1 +/- 12 Gg C yr^-1 for the five years without defoliation".*

Response: We have updated the manuscript (P. 19, L. 18-20): "In the study area of 100 $km^2$ the results suggested an average

40 decrease in regional GPP over the three outbreak years (2004, 2012, and 2013) of $15\pm5$ Gg C $yr^{-1}$, compared to the mean regional GPP of $40\pm12$ Gg C $yr^{-1}$ for the five years without defoliation."

*15. P2, L1. Reference(s) should support the "a warmer climate is likely to increase forest productivity" part of the sentence. Here are some suggestions: Pastor and Post (1988), Nemani et al. (2003), or Boisvenue and Running (2006).*

5     Response: We have added (Nemani et al. 2003; Boisvenue & Running 2006) as suggested (P. 2, L. 6).

*16. P2, L21. Since Brown et al. (2012) is already cited later on, I suggest mentioning it here too.*
Response: We have added (Brown et al. 2012) as suggested (P. 2, L. 26).

*17. P3, L3. Please give the temporal coverage of Landsat.*

10     Response: We have added the temporal coverage of Landsat (P. 3, L. 9):

*18. P3, L13. It is Monteith and Moss (1977).*

Response: We have updated the manuscript accordingly (P. 3, L. 19):

*19. P3, L27-28. Bright et al. (2013) already used remote sensing (Landsat to identify*
15 *trees killed by the mountain pine beetle) and a LUE model (MODIS GPP results, which are based on LUE) to quantify the impact of an insect outbreak on carbon uptake. To my knowledge, the authors can still claim being the first ones to do it for defoliators.*

Response: We have updated the manuscript accordingly and added Bright et al. (2013) as a reference (P. 3, L. 25-28).

20 *20. P3, L30-34. Delete the "This combination of [...]" sentence or merge it with the previous one (it is repetitive). Delete the "The method was developed [...]" sentence (it is repetitive with "Our main study objective [...]").*

Response: We have rewritten the section and the sentences are removed.

25 *21. P3, L34 to P4, L2. Delete or merge with the Methods.*

Response: The section is removed.

*22. P4, L2-4. The sentence is confusing, as it seems to imply that results will be provided for every year "between 2000 and 2015". I suggest re-writing along the following*
30 *lines: "Our main study objective was to compare GPP for years with (2004, 2012, and 2013) and without (2007, 2009, 2010, 2011, and 2014) insect outbreak in the birch forest of a subarctic valley of northern Sweden".*

Response: We have updated the manuscript accordingly (P. 3, L. 30-32): "Our main study objective was to compare GPP for
35 years with (2004, 2012 and 2013) and without (2007, 2009, 2010, 2011 and 2014) insect outbreak in the birch forest of a subarctic valley of northern Sweden."

*23. P5, L18 to P6, L1. The purpose of "quality classes based on QA data" is not clear: how where these classes used in the analysis?*

40     Response: We agree that mentioning the quality classes was confusing, hence, we have changed the text (P. 15, L. 17-18): "Double logistic functions were used to smooth the raw NDVI data and QA data from both MOD09Q1 and the more comprehensive QA-flags in MOD09A1 were utilized to estimate the quality of the NDVI observations."

*24. P6, L4. Replace "fraction canopy absorbed PAR" by "fraction of absorbed PAR by*

*the canopy". Similar comment for "canopy absorbed PAR" in Figure 4 caption.*

Response: We have updated the manuscript accordingly (P. 6, L. 6-7; Fig. 4, P. 14).

*25. P6, L8. Is the "pure canopy absorbed PAR" different from fAPAR_canopy? If yes, quickly explain what is the difference and why it matters. If not, replace the expression by "fAPAR_canopy".*

Response: Pure canopy absorbed PAR is the same as $fAPAR_{canopy}$. We have updated the manuscript accordingly (P. 6, L. 8).

*26. P6, L11. I would provide here (after "NDVI_DL") — instead of on P3, L20 — the reference to Myneni & Williams (1994) that supports the linear relationship.*

Response: We have updated the manuscript accordingly (P. 6, L. 12).

*27. P7. I suggest combining Figure 2 with Figure 1. For all multi-panel Figures, please add letters to each panel and refer to the appropriate panel in the text (e.g., Fig. 5a).*

Response: We do agree that it could be a good idea to merge the figures so that the orthophoto over the area around the EC-tower is shown together with the location of the tower. However, we decided to keep them as separate figures with the main motivation that we want to keep the size of Figure 2 large to make the photo easy to interpret. Letters have been added to all multi-panel figures.

*28. P8, L28. Please put here (instead of on P11, L8-10) the sentence about the value of the temperature lapse rate.*

Response: We have updated the manuscript accordingly (P. 8, L. 31-33).

*29. P9, L2. The units for PAR_8day should be MJ mˆ-2 day-1.*

Response: We have updated the manuscript accordingly (P. 9, L. 6).

*30. P9, L5. Add something like "(see Section 2.3.3)" so that readers know the value of epsilon_max will be discussed later. Similar comments for GDD_thresh (P9, L16) and T_thresh (P10, L2).*

Response: We have updated the manuscript accordingly (P. 9, L. 11; P. 9, L. 25; P. 10, L8).

*31. P9, Equation (7). Replace the current condition on the middle line by "-8 _ T_min8 < -3".*

Response: We have updated the manuscript accordingly (Eq. 7, P. 10).

*32. P10, L3. T_mean8 was never negative for such a northern site at the end of September? If negative values occurred, explain how they were dealt with (Equation (8) suggests a negative value for f_8day, which would make no sense).*

Response: There were no negative values for $T_{mean8}$ in the period studied (coldest value was 3°C). There were eight day periods with temperatures below zero, but no eight day period with a mean value $< 0°C$.

*33. P10, L6. Here and in Figure 5, replace "RMS" by "RMSE".*

Response: We have updated the manuscript accordingly (P. 10, L. 12; Fig. 5, P. 13).

*34. P10, Equation (9). Replace "nepsilon" by "nepsilon_max".*

Response: We have updated the manuscript accordingly (Eq. 9, P. 10).

*35. P11, L8. Delete the first sentence (repetitive with the previous paragraph).*

Response: We have removed the sentence.

*36. P11, L15-16. The text should be expanded, because at this point the reader is still unaware that an outbreak occurred in 2012 within the EC tower footprint: please state this explicitly here and add what was the percentage of defoliation within the tower footprint.*

Response: We have clarified according to the comment (P. 11, L. 21-23): "Two methods were applied to study the reduction in annual GPP due to the insect outbreaks: (1) a method based on a reduction factor derived from the EC data from 2012 when the birch forest in the footprint of the tower was severely defoliated, and no refoliation occurred."

*37. P12, L1-2. The sentence is cumbersome; I suggest re-writing along the following lines: "For each year with insect outbreak, the regional reduction in GPP was computed by summing, over all pixels identified as defoliated, the difference between the mean GPP for no-outbreak years and the GPP for this specific outbreak year".*

Response: We have updated according to the suggestion (P. 11, L. 10-11).

*38. P13, L6. For consistency, replace "GDD threshold" by "GDD_thresh", and "temperature factor" by "T_thresh".*

Response: We have updated according to the suggestion (P. 13, L. 6).

*39. P14, Figure 6 legend. For consistency, replace "f_MOD8" by "f_8day".*

Response: We have updated according to the suggestion (Fig. 6, P. 14).

*40. P15, L8. Here and throughout the text (including Figures): provide full units for local, mean, and regional GPP (i.e., with yrˆ-1) even when writing "annual".*

Response: We have updated the manuscript according to the comment.

*41. P16, L14-15. This sentence is repetitive with P11, L4-6.*

Response: We have removed the sentence.

*42. P17, L1-2. Make it clearer that the 41.1 value was computed as the mean over nooutbreak years, based on the GPP_lue equation (add a number) given on P14. Please also add this information as a footnote to Table 2.*

Response: We have clarified according to the comment (P. 17, L. 6): "The average annual regional GPP in the study area, derived with the LUE model (Eq. 12) and the five years without insect outbreak…" and added a footnote to Table 2.

*43. P18, Figure 9 caption. Replace "birch moth outbreaks" by "outbreaks of autumnal moth and winter moth".*

Response: The figue caption is updated according to the comment.

*44. P19, L2. The word "demonstrated" seems too strong; similar comment for P21, L17.*

Response: We have changed to shown (P. 19, L. 5) and (P.23, L. 31).

*45. P19, L3. Replace "decreased with 260 g C mˆ-2" by "decreased by 261 g C mˆ-2 yrˆ-1" (based on Table 1, the difference is 261). Similar comment for P20, L25.*

Response: We have changed all values to two significant digits; hence, this comment is no longer relevant.

*46. P19, L5. According to Table 2, the highest value is 265 +/- 93 (not 244 +/- 73).*

Response: Thanks for noting. We have updated the manuscript accordingly (P. 19, L. 4).

*47. P20, L31. Replace "turned a forest into a carbon sink" by "turned a forest into a carbon source". Stands were carbon sinks only during the growing season; the main point here is that insect-caused mortality led to negative annual NEP... although these stands seem to be recovering quickly, as correctly noted by the authors in the second part of the sentence!*

Response: Thanks for noting. This was a mistake that has been corrected now.

*48. P21, L1-2. For (partially) contrasting modeling results about the effect of insect outbreaks on the carbon balance, see Seidl et al. (2008), Albani et al. (2010), and Landry et al. (2016) (the last one is not the same reference as already cited in the manuscript).*

Response: Thanks for these interesting references. We did not include them though as they mainly discuss longer term impacts.

*49. P21, L10-11. Delete the whole "since it has been suggested [...] (Medvigy et al. 2012)" part of the sentence. The "spatial distribution of defoliation" issue addressed by Medvigy et al. (2012) dealt with tree-level defoliation within a stand (e.g., 100% defoliation of 40% of trees vs. 40% defoliation of 100% of trees). This is not something addressed here, nor is MODIS-like remote sensing appropriate for this.*

Response: We have removed the part of the sentence as suggested.

*50. P21, L13-14. Delete the sentence: there is no firm basis for such an extrapolation, and the number would then likely end up being cited by future studies...*

Response: Since these insect infestations occur frequently across very large areas in the region we think that it is relevant to include some information about the potential effect on the carbon uptake. However, we agree that it is an uncertain estimation and reformulated the text accordingly (P. 23, L. 27-29): "Assuming that the conditions were similar over northern Fennoscandia, the insect defoliation over these vast areas would result in a potential total regional GPP loss for the time period of the magnitude 2–3 Tg C. Models not accounting for such recurring disturbance events would seriously overestimate the ability of these forests to absorb atmospheric $CO_2$."

| | Years without insect outbreak | | | | | Outbreak |
|---|---|---|---|---|---|---|
| Year | 2007 | 2009 | 2010 | 2011 | 2014 | 2012 |
| **GPP** (g C m$^{-2}$ yr$^{-1}$) | 450 | 530 | 370 | 400 | 450 | 180 |

where annual GPP was largest in 2007 and 2009, and the lower annual GPP in 2010 and 2011 indicates that there were minor defoliation already in these years.

25 Due to the limited number of years with EC derived GPP we did not consider it reliable to study correlations between annual GPP measured at the EC tower and meteorological variables. Instead we modeled GPP for the birch forest around the tower with PAR, and compared EC derived and PAR-modeled GPP. The comparison suggests that in the two years (2010 and 2011) prior to the outbreak, measured GPP was lower than PAR-modeled GPP, indicating that there were signs of defoliation by growing larval densities. Also in 2014 when the birch forest most likely was recovering from the outbreak,
30 measured GPP was lower than PAR-modeled GPP. For the earlier years (2007 and 2009) when the birch forest was likely closer to undisturbed conditions, EC derived GPP and GPP modeled with PAR data agreed well; measured GPP was slightly lower compared to PAR-modeled in 2007 and slightly larger in 2009. For the outbreak year 2012 the difference between EC derived GPP and PAR-modeled GPP was 290 g C m$^{-2}$ yr$^{-1}$, which is similar to the decrease of 260 g C m$^{-2}$ yr$^{-1}$ we found in our study. In addition, we ran the LUE model with meteorological data from the scientific research station in Abisko (ANS)
35 for the year 2008 to fill the gap in the time-series with measured GPP and to study how well it agreed with the years 2007

and 2009. According to the LUE model the annual GPP at the EC tower was 440 g C $m^{-2}$ $yr^{-1}$ in 2008. This indicates that the GPP for undisturbed years of 440 g C $m^{-2}$ $yr^{-1}$ that we used is reasonable. However, with data from the EC tower available for more years it would be a potentially important improvement to include meteorological data when estimating the decrease in annual GPP. We have added a section about these results in the discussion (P. 20, L. 7-20) and the figure showing EC derived and PAR-modeled GPP was added to the supplementary material.

We also studied correlations between NDVI and meteorological data available from ANS, where we used mean of the highest seasonal $NDVI_{DL}$ value derived from 200 MODIS pixels with birch forest. To minimize the influence of insect induced defoliation we excluded the outbreak years and years immediately prior to and after outbreaks. No linear relationships between PAR and GPP were found. There were, however, negative correlations between temperature and $NDVI_{DL}$, with the strongest correlation between $NDVI_{DL}$ and the mean temperature in May-June. The influence of the temperature on NDVI was however, weak and due the large estimated uncertainties of the LUE model (30%) we did not include these correlations in the analysis. We do, however, mention these results in the discussion (P. 20, L. 21-32) but due to the limited amount of data we do not further elaborate on the results as that would be speculation.

*The comparison with the 2004 results of Heliasz et al. (2011) on page 20 is useful and indicates that closer analysis of the temporal evolution of the EC data may be beneficial. Currently data are separated into years with and without insect outbreaks. During those years with outbreaks, does the reduction in GPP over the course of the growing season agree with the timing of insect population growth/insect damage? Is this also supported by the NDVI data? In the years classified as being without insect outbreaks, are there any effects of (albeit smaller) insect populations on the GPP or NDVI values?*

*Perhaps such analyses could offer further insight into the refoliation effect; it is currently hard to draw meaningful conclusions on this subject from the information given in the article. Some evidence to support the assumption that NDVIDL captures refoliation would also be useful; Fig 3 is not very convincing in this respect.*

Response: For the year 2004, when we had EC data for the later part of the insect defoliation and the following refoliation at the EC tower, we can see that GPP is low during the defoliation events and increasing later in the growing season with the new leaves appearing. We have clarified this in the discussion (P. 22, L. 5-17).

As mentioned in our previous comment we found influence on GPP at the EC tower the two years prior to the outbreak that are likely due to increasing insect population before reaching outbreak levels. This pattern is suggested in time-series of NDVI where there seems to be weak signs of defoliation 1-2 years prior to the outbreak.

*P15, L6-10: Section 2.2.3 states that EC data were available from 1 May to 30 Sep, covering most of the growing season. Do the EC observations and values given in Table 1 agree with this timeframe, or is it possible GPP in Table 1 is underestimated if the growing season extended beyond these dates? This study focuses on GPP, but could the authors comment on other potential impacts of the insect outbreak on the carbon balance? For example, how might respiration rates be affected, and might this impact the partitioning into Reco and GPP?*

Response: Budburst usually occurs early in June or very late in May so including data from first of May means that we capture the start of the growing season. For the years included in this study GPP was approaching zero by the last week of September implying that there were no underestimates of annual GPP. We have clarified this (P. 8, L. 5-10).

We have also added a discussion about respiration (P. 19, L. 11-18): " In this study we have estimated the impact on GPP only but we noted that during the outbreak in 2012 the decrease in $R_{eco}$ was larger than the decrease in GPP during the growing season around the EC tower. Respiration is affected by insect outbreak in two ways: (1) Autotrophic respiration is reduced as defoliated trees cannot photosynthesize, and (2) heterotrophic respiration increases when dead larvae decompose. The amount of carbon respired by larvae is likely to be the same as the amount of carbon in eaten leaves, so we should only observe a shift of respiration in time. In addition, larvae transport nutrients from trees to fungi and bacteria living in soil, which further increase respiration. The increase in heterotrophic respiration did not offset decrease in autotrophic respiration, and $R_{eco}$ for outbreak year was decreased in comparison to non-disturbed years."

*On a similar note, many of the decisions taken in the presentation of results and development of the model rely on data collected during non-outbreak years. Is the gap-filling approach also suitable in defoliated years?*

Response: The gap-filling approach is suitable also for defoliated years. We have clarified this in the manuscript (P. 8, L. 15-17): "We considered the gap-filling approach suitable also for defoliated years since the gap-filling function is created based on data from short time windows, usually seven days, and hence, do adjust the fitting parameters for changing ecosystem conditions."

*P8, L10-11: More detail is needed about the quality control. Under which 'bad' weather/measuring conditions were data removed? How much data remained after quality control and what proportion was gap-filled?*

Response: Data were removed mainly due to precipitation since an open path gas analyzer was used, or if the atmosphere was not fulfilling the turbulent conditions required for eddy covariance measurements. Available data after cleaning: 2007 61%; 2009 71%; 2010 66%; 2011 65%; 2012 58%; 2014 65%. We have clarified this in the manuscript (P. 8, L. 13-15): "The main reasons for removing data were precipitation as we used an open path gas analyser, and atmospheric conditions that did not fulfill turbulent the conditions required."

*P10, L11-2: Here, it is not clear which years have been used or why. The years for which the EC data are available should be stated in Section 2.2.3. Why was 2012 the only year used to calculate "max with insect defoliation? Why were data from 2008 and 2013 not included/not available?*

Response: We have clarified this in Section 2.3.3 (P. 10, L. 18-21): "Two $\varepsilon_{max}$ values were computed: one including data from the five years (2007, 2009, 2010, 2011 and 2014) with undisturbed birch forest, and one ($\varepsilon_{max, def}$) for the year 2012 with insect defoliation. No data were available from 2008 due to equipment failure, and in 2013 the measurements were disturbed by larvae climbing the equipment."

*P12, L13: It is not clear how these statistics were calculated and they don't seem to follow from Fig 4. Please provide more details/clarification*

Response: We do agree that these statistics were not well described. Previously the statistics were based on the entire study area. We have changed the statistics to include only the pixels around the EC tower to correspond to the figure and clarified this (P. 12, L. 22-24): "The influence of observations with $NDVI_{DL}$ values $< 0.4$ and with $f_{8day} > 0$ was small. For the years

with data available from the EC tower 8% of the eight day periods had $NDVI_{DL} < 0.4$ and $f_{8day} > 0$ in the MODIS pixels surrounding the tower. For these time periods average $f_{8day}$ was 0.068.

*P14, Fig 6: The two green lines for NDVIDL show higher NDVI values early in the growing season for the defoliated year. Possible reasons for this should be investigated and the impact on the results commented on.*

Response: Thanks for pointing this out. This is due to a weak fitting of the double logistic functions in TIMESAT. In a currently developed version of TIMESAT the fitting of the functions will be more robust. We have added a clarification in Section 3.3.1 (P. 16, L. 7-10):"In Figure 6, $NDVI_{DL}$ has higher values in the year with defoliation compared to undisturbed years in May (period 16-18). These high $NDVI_{DL}$ values are due to poor fitting of the double logistic function during winter and early spring in 2012 (see Figure 3, where $NDVI_{DL}$ increases earlier in 2012 compared to the other years). The impact on the result is however small since these eight periods are in the early part of the growing season when $f_{8day}$ is zero.

**Minor comments:**
*P2, L22: Change 'difference' to 'differences'*

Response: We have updated the manuscript accordingly.

*P2, L26: Would be useful to give the land cover type for the southern France site*

Response: We have added that the defoliation occurred in holm oak (Quercus ilex L.) (P. 2, L. 31).

*P3, L17: Change to read 'of the form'*
*P3, L18: Number this equation and update the others accordingly*
*P3, L20-1: Change to read 'and with variability in meteorology'*

Response: The section is updated and these comments are no longer relevant.

*P4, L24-6: It is not clear that these recent outbreaks are for the study site – please state the area they apply to. Also give some information about the EC tower in that study (i.e. location, and mention the flux measurements were also for the birch forest)*

Response: We have clarified that the outbreaks mentioned were in the study area, and that the EC tower mentioned was located in birch forest (P. 4, L. 17-19): "The latest outbreaks in the study are occurred in 2004, with a documented reduction in carbon sink strength of 89% at an EC tower located in birch forest…."

*P5, L10: Change 'derived' to 'derive'*

Response: We have updated the manuscript accordingly.

*P6, L16-7: Change to read 'to be about'*

Response: We have updated the manuscript accordingly.

*P6, L18: Keep the order of 'east/west' the same as previously (L17). When the wind is from the west, the footprint is located to the west of the tower*

Response: We have updated the manuscript accordingly (P. 6, L. 18-20): "The prevailing wind directions are from the west and from the east, hence the main footprint of the EC tower is to the west and east from the tower where vegetation is most homogenous."

P6, L24: Change 'wind speeds' to 'stability', as wind direction and stability tend to be the major controls on EC footprints

Response: We have updated the manuscript accordingly.

P7, Fig 3: Change y-axis of RH plot to 'NDVIDL'

Response: We have updated the manuscript accordingly.

P8, L16: Change 'measured the' to 'the measured'

Response: We have updated the manuscript accordingly.

P9, Eq3: Units on LHS do not equate to units on RHS

Response: We have updated the manuscript accordingly.

P11, L22-23 Suggest choosing alternative notation for GPPreduction, as it is a ratio of GPPs, rather than GPP itself

Response: We have changed the notation of the reduction factor to *GPPredfact*.

P12, L6: Change 'accurate' to 'accurately'

Response: We have updated the manuscript accordingly.

P13, Fig 5: Add units for RMS

Response: We have updated the manuscript accordingly and also changed RMS to RMSE (Fig. 5, P. 13).

P14 L5-6 and Fig 7: The text mentions 'low GPP observations' but in Fig 7 it looks as though the modelled GPP values are lower than the EC observed values, with several zero values. Please clarify

[revised manuscript text omitted]